# A statistical framework to assess cross-frequency coupling while accounting for confounding analysis effects

Jessica K Nadalin[1], Louis-Emmanuel Martinet[2], Ethan B Blackwood[3], Meng-Chen Lo[3], Alik S Widge[3], Sydney S Cash[2], Uri T Eden[1], Mark A Kramer[1]*

[1]Department of Mathematics and Statistics, Boston University, Boston, United States; [2]Department of Neurology, Massachusetts General Hospital, Boston, United States; [3]Department of Psychiatry, University of Minnesota, Minneapolis, United States

**Abstract** Cross frequency coupling (CFC) is emerging as a fundamental feature of brain activity, correlated with brain function and dysfunction. Many different types of CFC have been identified through application of numerous data analysis methods, each developed to characterize a specific CFC type. Choosing an inappropriate method weakens statistical power and introduces opportunities for confounding effects. To address this, we propose a statistical modeling framework to estimate high frequency amplitude as a function of both the low frequency amplitude and low frequency phase; the result is a measure of phase-amplitude coupling that accounts for changes in the low frequency amplitude. We show in simulations that the proposed method successfully detects CFC between the low frequency phase or amplitude and the high frequency amplitude, and outperforms an existing method in biologically-motivated examples. Applying the method to in vivo data, we illustrate examples of CFC during a seizure and in response to electrical stimuli.

DOI: https://doi.org/10.7554/eLife.44287.001

*For correspondence:
mak@bu.edu

**Competing interests:** The authors declare that no competing interests exist.

## Introduction

Brain rhythms - as recorded in the local field potential (LFP) or scalp electroencephalogram (EEG) - are believed to play a critical role in coordinating brain networks. By modulating neural excitability, these rhythmic fluctuations provide an effective means to control the timing of neuronal firing (*Engel et al., 2001*; *Buzsáki and Draguhn, 2004*). Oscillatory rhythms have been categorized into different frequency bands (e.g., theta [4–10 Hz], gamma [30–80 Hz]) and associated with many functions: the theta band with memory, plasticity, and navigation (*Engel et al., 2001*); the gamma band with local coupling and competition (*Kopell et al., 2000*; *Börgers et al., 2008*). In addition, gamma and high-gamma (80–200 Hz) activity have been identified as surrogate markers of neuronal firing (*Rasch et al., 2008*; *Mukamel et al., 2005*; *Fries et al., 2001*; *Pesaran et al., 2002*; *Whittingstall and Logothetis, 2009*; *Ray and Maunsell, 2011*), observable in the EEG and LFP.

In general, lower frequency rhythms engage larger brain areas and modulate spatially localized fast activity (*Bragin et al., 1995*; *Chrobak and Buzsáki, 1998*; *von Stein and Sarnthein, 2000*; *Lakatos et al., 2005*; *Lakatos et al., 2008*). For example, the phase of low frequency rhythms has been shown to modulate and coordinate neural spiking (*Vinck et al., 2010*; *Hyafil et al., 2015b*; *Fries et al., 2007*) via local circuit mechanisms that provide discrete windows of increased excitability. This interaction, in which fast activity is coupled to slower rhythms, is a common type of cross-frequency coupling (CFC). This particular type of CFC has been shown to carry behaviorally relevant information (e.g., related to position [*Jensen and Lisman, 2000*; *Agarwal et al., 2014*], memory

[*Siegel et al., 2009*], decision making and coordination [*Dean et al., 2012*; *Pesaran et al., 2008*; *Wong et al., 2016*; *Hawellek et al., 2016*]). More generally, CFC has been observed in many brain areas (*Bragin et al., 1995*; *Chrobak and Buzsáki, 1998*; *Csicsvari et al., 2003*; *Tort et al., 2008*; *Mormann et al., 2005*; *Canolty et al., 2006*), and linked to specific circuit and dynamical mechanisms (*Hyafil et al., 2015b*). The degree of CFC in those areas has been linked to working memory, neuronal computation, communication, learning and emotion (*Tort et al., 2009*; *Jensen et al., 2016*; *Canolty and Knight, 2010*; *Dejean et al., 2016*; *Karalis et al., 2016*; *Likhtik et al., 2014*; *Jones and Wilson, 2005*; *Lisman, 2005*; *Sirota et al., 2008*), and clinical disorders (*Gordon, 2016*; *Widge et al., 2017*; *Voytek and Knight, 2015*; *Başar et al., 2016*; *Mathalon and Sohal, 2015*), including epilepsy (*Weiss et al., 2015*). Although the cellular mechanisms giving rise to some neural rhythms are relatively well understood (e.g. gamma [*Whittington et al., 2000*; *Whittington et al., 2011*; *Mann and Mody, 2010*]), the neuronal substrate of CFC itself remains obscure.

Analysis of CFC focuses on relationships between the amplitude, phase, and frequency of two rhythms from different frequency bands. The notion of CFC, therefore, subsumes more specific types of coupling, including: phase-phase coupling (PPC), phase-amplitude coupling (PAC), and amplitude-amplitude coupling (AAC) (*Hyafil et al., 2015b*). PAC has been observed in rodent striatum and hippocampus (*Tort et al., 2008*) and human cortex (*Canolty et al., 2006*), AAC has been observed between the alpha and gamma rhythms in dorsal and ventral cortices (*Popov et al., 2018*), and between theta and gamma rhythms during spatial navigation (*Shirvalkar et al., 2010*), and both PAC and AAC have been observed between alpha and gamma rhythms (*Osipova et al., 2008*). Many quantitative measures exist to characterize different types of CFC, including: mean vector length or modulation index (*Canolty et al., 2006*; *Tort et al., 2010*), phase-locking value (*Mormann et al., 2005*; *Lachaux et al., 1999*; *Vanhatalo et al., 2004*), envelope-to-signal correlation (*Bruns and Eckhorn, 2004*), analysis of amplitude spectra (*Cohen, 2008*), coherence between amplitude and signal (*Colgin et al., 2009*), coherence between the time course of power and signal (*Osipova et al., 2008*), and eigendecomposition of multichannel covariance matrices (*Cohen, 2017*). Overall, these different measures have been developed from different principles and made suitable for different purposes, as shown in comparative studies (*Tort et al., 2010*; *Cohen, 2008*; *Penny et al., 2008*; *Onslow et al., 2011*).

Despite the richness of this methodological toolbox, it has limitations. For example, because each method focuses on one type of CFC, the choice of method restricts the type of CFC detectable in data. Applying a method to detect PAC in data with both PAC and AAC may: (i) falsely report no PAC in the data, or (ii) miss the presence of significant AAC in the same data. Changes in the low frequency power can also affect measures of PAC; increases in low frequency power can increase the signal to noise ratio of phase and amplitude variables, increasing the measure of PAC, even when the phase-amplitude coupling remains constant (*Aru et al., 2015*; *van Wijk et al., 2015*; *Jensen et al., 2016*). Furthermore, many experimental or clinical factors (e.g., stimulation parameters, age or sex of subject) can impact CFC in ways that are difficult to characterize with existing methods (*Cole and Voytek, 2017*). These observations suggest that an accurate measure of PAC would control for confounding variables, including the power of low frequency oscillations.

To that end, we propose here a generalized linear model (GLM) framework to assess CFC between the high-frequency amplitude and, simultaneously, the low frequency phase and amplitude. This formal statistical inference framework builds upon previous work (*Kramer and Eden, 2013*; *Penny et al., 2008*; *Voytek et al., 2013*; *van Wijk et al., 2015*) to address the limitations of existing CFC measures. In what follows, we show that this framework successfully detects CFC in simulated signals. We compare this method to the modulation index, and show that in signals with CFC dependent on the low-frequency amplitude, the proposed method more accurately detects PAC than the modulation index. We apply this framework to in vivo recordings from human and rodent cortex to show examples of PAC and AAC detected in real data, and how to incorporate new covariates directly into the model framework.

## Materials and methods

### Estimation of the phase and amplitude envelope

To study CFC we estimate three quantities: the phase of the low frequency signal, $\phi_{\text{low}}$; the amplitude envelope of the high frequency signal, $A_{\text{high}}$; and the amplitude envelope of the low frequency signal, $A_{\text{low}}$. To do so, we first bandpass filter the data into low frequency (4–7 Hz) and high frequency (100–140 Hz) signals, $V_{\text{low}}$ and $V_{\text{high}}$, respectively, using a least-squares linear-phase FIR filter of order 375 for the high frequency signal, and order 50 for the low frequency signal. Here we choose specific high and low frequency ranges of interest, motivated by previous in vivo observations (*Canolty et al., 2006*; *Tort et al., 2008*; *Scheffer-Teixeira et al., 2013*). However, we note that this method is flexible and not dependent on this choice. We select a wide high frequency band consistent with recommendations from the literature (*Aru et al., 2015*) and the mechanistic explanation that extracellular spikes produce this broadband high frequency activity (*Scheffer-Teixeira et al., 2013*). We use the Hilbert transform to compute the analytic signals of $V_{\text{low}}$ and $V_{\text{high}}$, and from these compute the phase and amplitude of the low frequency signal ($A_{\text{low}}$ and $\phi_{\text{low}}$) and the amplitude of the high frequency signal ($A_{\text{high}}$).

### Modeling framework to assess CFC

Generalized linear models (GLMs) provide a principled framework to assess CFC (*Penny et al., 2008*; *Kramer and Eden, 2013*; *van Wijk et al., 2015*). Here, we present three models to analyze different types of CFC. The fundamental logic behind this approach is to model the distribution of $A_{\text{high}}$ as a function of different predictors. In existing measures of PAC, the distribution of $A_{\text{high}}$ versus $\phi_{\text{low}}$ is assessed using a variety of different metrics (e.g., *Tort et al., 2010*). Here, we estimate statistical models to fit $A_{\text{high}}$ as a function of $\phi_{\text{low}}$, $A_{\text{low}}$, and their combinations. If these models fit the data sufficiently well, then we estimate distances between the modeled surfaces to measure the impact of each predictor.

### The $\phi_{\text{low}}$ model

The $\phi_{low}$ *model* relates $A_{\text{high}}$, the response variable, to a linear combination of $\phi_{\text{low}}$, the predictor variable, expressed in a spline basis:

$$A_{\text{high}} | \phi_{\text{low}} \sim \text{Gamma}[\mu, \nu] \tag{1}$$

$$\log \mu = \sum_{k=1}^{n} \beta_k f_k(\phi_{\text{low}}),$$

where the conditional distribution of $A_{\text{high}}$ given $\phi_{\text{low}}$ is modeled as a Gamma random variable with mean parameter $\mu$ and shape parameter $\nu$, and $\beta_k$ are undetermined coefficients, which we refer to collectively as $\beta_{\phi_{\text{low}}}$. We choose this distribution as it guarantees real, positive amplitude values; we note that this distribution provides an acceptable fit to the example human data analyzed here (*Figure 1*). The functions $\{f_1, \cdots, f_n\}$ correspond to spline basis functions, with $n$ control points equally spaced between 0 and $2\pi$, used to approximate $\phi_{\text{low}}$. We note that the spline functions sum to 1, and therefore we omit a constant offset term. We use a tension parameter of 0.5, which controls the smoothness of the splines. We note that, because the link function of the conditional mean of the response variable ($A_{\text{high}}$) varies linearly with the model coefficients $\beta_k$ the model is a GLM, though the spline basis functions situate the model in the larger class of Generalized Additive Models (GAMs). Here we fix $n = 10$, which is a reasonable choice for smooth PAC with one or two broad peaks (*Kramer and Eden, 2013*). To support this choice, we apply an AIC-based selection procedure to 1000 simulated instances of signals of duration 20 s with phase-amplitude coupling and amplitude-amplitude coupling (see Materials and methods: *Synthetic Time Series with PAC* and *Synthetic Time Series with AAC*, below, for simulation details). For each simulation, we fit the model in *Equation 1* to these data for 27 different values of $n$ from $n = 4$ to $n = 30$. For each simulated signal, we record the value of $n$ that minimizes the AIC, defined as

$$\text{AIC} = \Delta + 2n,$$

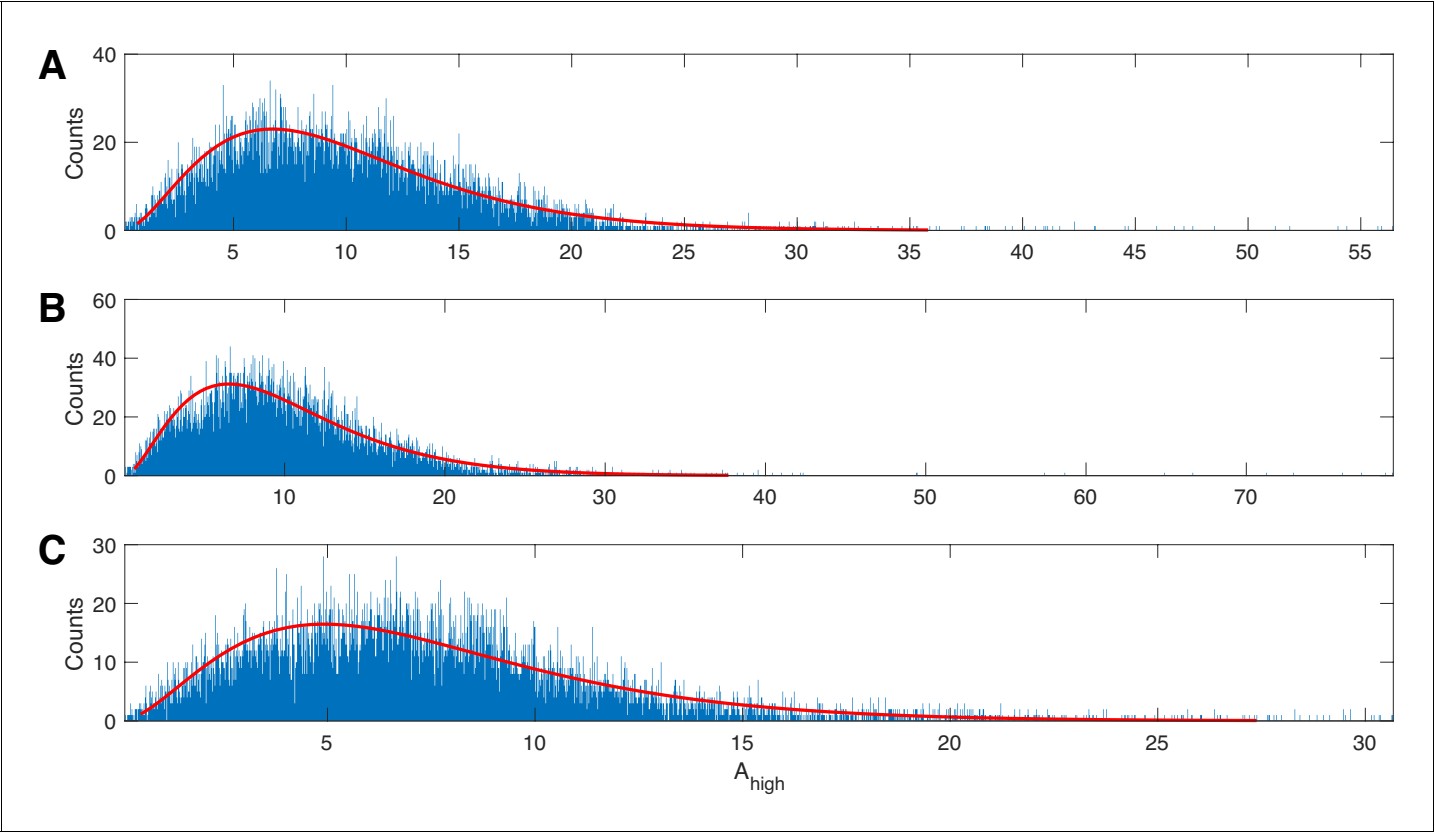

**Figure 1.** The gamma distribution provides a good fit to example human data. Three examples of 20 s duration recorded from a single electrode during a human seizure. In each case, the gamma fit (red curve) provides an acceptable fit to the empirical distributions of the high frequency amplitude.

DOI: https://doi.org/10.7554/eLife.44287.002

where $\Delta$ is the deviance from the model in *Equation 1*. The values of $n$ that minimize the AIC tend to lie between $n = 7$ and $n = 12$ (*Figure 2*). These simulations support the choice of $n = 10$ as a sufficient number of splines.

For a more detailed discussion and simulation examples of the PAC model, see *Kramer and Eden (2013)*. We note that the choices of distribution and link function differ from those in *Penny et al. (2008)* and *van Wijk et al. (2015)*, where the normal distribution and identity link are used instead.

### The $A_{\text{low}}$ model

The $A_{low}$ *model* relates the high frequency amplitude to the low frequency amplitude:

$$A_{\text{high}}|A_{\text{low}} \sim \text{Gamma}[\mu, \nu] \tag{2}$$

$$\log \mu = \beta_1 + \beta_2 A_{\text{low}},$$

where the conditional distribution of $A_{\text{high}}$ given $A_{\text{low}}$ is modeled as a Gamma random variable with mean parameter $\mu$ and shape parameter $\nu$. The predictor consists of a single variable and a constant, and the length of the coefficient vector $\beta_{A_{\text{low}}} = \{\beta_1, \beta_2\}$ is 2.

### The $A_{\text{low}}, \phi_{\text{low}}$ model

The $A_{low}, \phi_{low}$ *model* extends the $\phi_{low}$ model in *Equation 1* by including three additional predictors in the GLM: $A_{\text{low}}$, the low frequency amplitude; and interaction terms between the low frequency amplitude and the low frequency phase: $A_{\text{low}} \sin(\phi_{\text{low}})$, and $A_{\text{low}} \cos(\phi_{\text{low}})$. These new terms allow

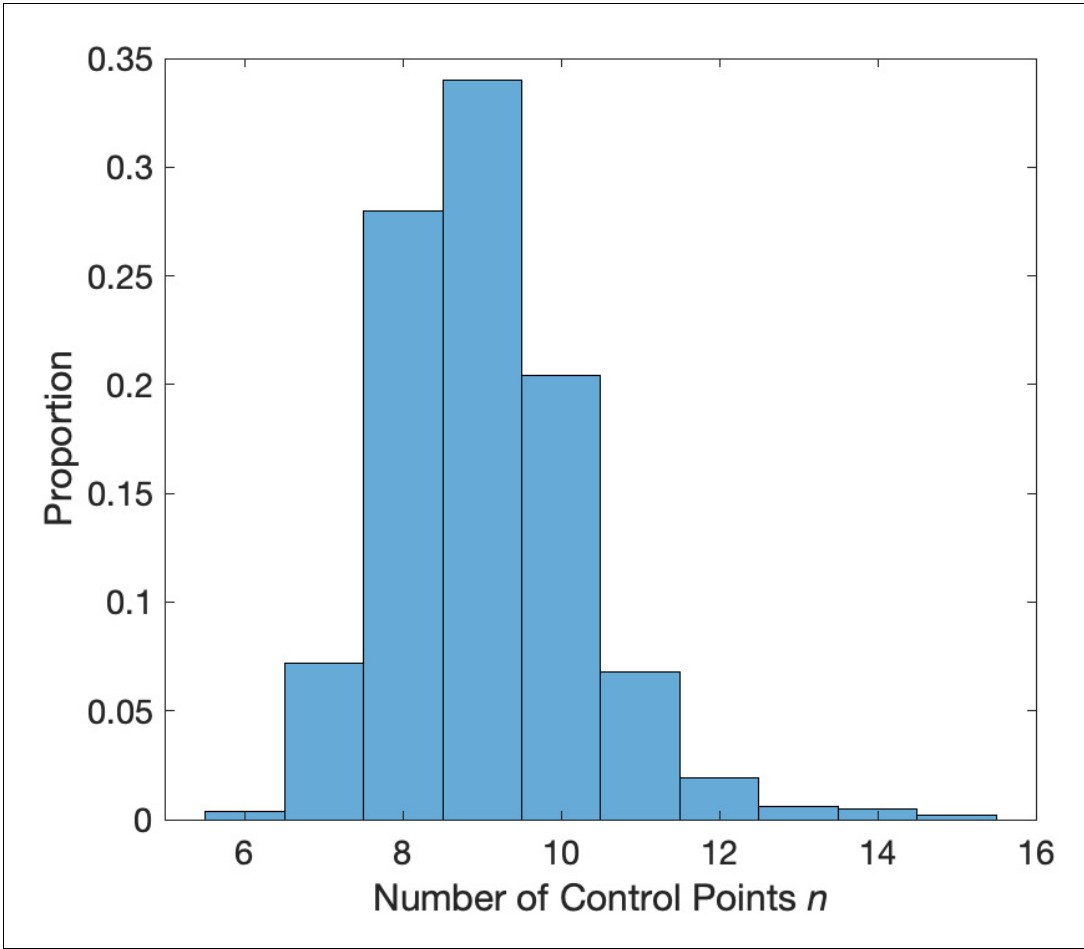

**Figure 2.** Distribution of the number of control points ($n$) that minimize the AIC. Values of $n$ between 7 and 12 minimize the AIC in a simulation with phase-amplitude coupling and amplitude-amplitude coupling.

DOI: https://doi.org/10.7554/eLife.44287.003

assessment of phase-amplitude coupling while accounting for linear amplitude-amplitude dependence and more complicated phase-dependent relationships on the low frequency amplitude without introducing many more parameters. Compared to the original $\phi_{\mathrm{low}}$ model in *Equation 1*, including these new terms increases the number of variables to $n+3$, and the length of the coefficient vector $\beta_{A_{\mathrm{low}},\phi_{\mathrm{low}}}$ to $n+3$. These changes result in the following model:

$$A_{\mathrm{high}}|\phi_{\mathrm{low}}, A_{\mathrm{low}} \sim \mathrm{Gamma}[\mu, \nu], \tag{3}$$

$$\log \mu = \sum_{k=1}^{n} \beta_k f_k(\phi_{\mathrm{low}}) + \beta_{n+1} A_{\mathrm{low}} + \beta_{n+2} A_{\mathrm{low}} \sin(\phi_{\mathrm{low}}) + \beta_{n+3} A_{\mathrm{low}} \cos(\phi_{\mathrm{low}}).$$

Here, the conditional distribution of $A_{\mathrm{high}}$ given $\phi_{\mathrm{low}}$ and $A_{\mathrm{low}}$ is modeled as a Gamma random variable with mean parameter $\mu$ and shape parameter $\nu$, and $\beta_k$ are undetermined coefficients. We note that we only consider two interaction terms, rather than the spline basis function of phase, to limit the number of parameters in the model.

### The statistics $\mathrm{R}_{\mathrm{PAC}}$ and $\mathrm{R}_{\mathrm{AAC}}$

We compute two measures of CFC, $\mathrm{R}_{\mathrm{PAC}}$ and $\mathrm{R}_{\mathrm{AAC}}$ which use the three models defined in the previous section. We evaluate each model in the three-dimensional space ($\phi_{\mathrm{low}}, A_{\mathrm{low}}, A_{\mathrm{high}}$) and calculate the statistics $\mathrm{R}_{\mathrm{PAC}}$ and $\mathrm{R}_{\mathrm{AAC}}$. We use the MATLAB (*RRID : SCR_0 01622*) function *fitglm* to estimate

the models; we note that this procedure estimates the dispersion directly for the gamma distribution. In what follows, we first discuss the three model surfaces estimated from the data, and then how we use these surfaces to compute the statistics $\mathbf{R}_{\mathrm{PAC}}$ and $\mathbf{R}_{\mathrm{AAC}}$.

To create the surface $S_{A_{\mathrm{low}},\phi_{\mathrm{low}}}$, which fits the $A_{\mathrm{low}},\phi_{\mathrm{low}}$ model in the three-dimensional ($A_{\mathrm{low}}$, $\phi_{\mathrm{low}}$, $A_{\mathrm{high}}$) space, we first compute estimates of the parameters $\beta_{A_{\mathrm{low}},\phi_{\mathrm{low}}}$ in *Equation 3*. We then estimate $A_{\mathrm{high}}$ by fixing $A_{\mathrm{low}}$ at one of 640 evenly spaced values between the 5th and 95th quantiles of $A_{\mathrm{low}}$ observed; we choose these quantiles to avoid extremely small or large values of $A_{\mathrm{low}}$. Finally, at the fixed $A_{\mathrm{low}}$, we compute the high frequency amplitude values from the $A_{\mathrm{low}},\phi_{\mathrm{low}}$ model over 100 evenly spaced values of $\phi_{\mathrm{low}}$ between $-\pi$ and $\pi$. This results in a two-dimensional curve $C_{A_{\mathrm{low}},\phi_{\mathrm{low}}}$ in the two-dimensional ($\phi_{\mathrm{low}}$, $A_{\mathrm{high}}$) space with fixed $A_{low}$. We repeat this procedure for all 640 values of $A_{low}$ to create a surface $S_{A_{\mathrm{low}},\phi_{\mathrm{low}}}$ in the three-dimensional space ($A_{\mathrm{low}}$, $\phi_{\mathrm{low}}$, $A_{\mathrm{high}}$) (*Figure 3C*). To create the surface $S_{A_{low}}$, which fits the $A_{\mathrm{low}}$ model in the three-dimensional ($A_{\mathrm{low}}$, $\phi_{\mathrm{low}}$, $A_{\mathrm{high}}$) space, we estimate the coefficient vector $\beta_{A_{\mathrm{low}}}$ for the model in *Equation 2*. We then estimate the high frequency amplitude over 640 evenly spaced values between the 5th and 95th quantiles of $A_{\mathrm{low}}$ observed, again to avoid extremely small or large values of $A_{\mathrm{low}}$. This creates a mean response function which appears as a curve $C_{A_{\mathrm{low}}}$ in the two-dimensional ($A_{\mathrm{low}}$, $A_{\mathrm{high}}$) space. We extend this two-dimensional curve to a three-dimensional surface $S_{A_{\mathrm{low}}}$ by extending $C_{A_{\mathrm{low}}}$ along the $\phi_{\mathrm{low}}$ dimension (*Figure 3A*).

To create the surface $S_{\phi_{\mathrm{low}}}$, which fits the $\phi_{\mathrm{low}}$ model in the three-dimensional ($A_{\mathrm{low}}$, $\phi_{\mathrm{low}}$, $A_{\mathrm{high}}$) space, we first estimate the coefficients $\beta_{\phi_{\mathrm{low}}}$ for the model in *Equation 1*. From this, we then compute estimates for the high frequency amplitude using the $\phi_{\mathrm{low}}$ model with 100 evenly spaced values of $\phi_{\mathrm{low}}$ between $-\pi$ and $\pi$. This results in the mean response function of the $\phi_{\mathrm{low}}$ model. We extend this curve $C_{\phi_{\mathrm{low}}}$ in the $A_{\mathrm{low}}$ dimension to create a surface $S_{\phi_{\mathrm{low}}}$ in the three-dimensional ($A_{\mathrm{low}}$, $\phi_{\mathrm{low}}$,

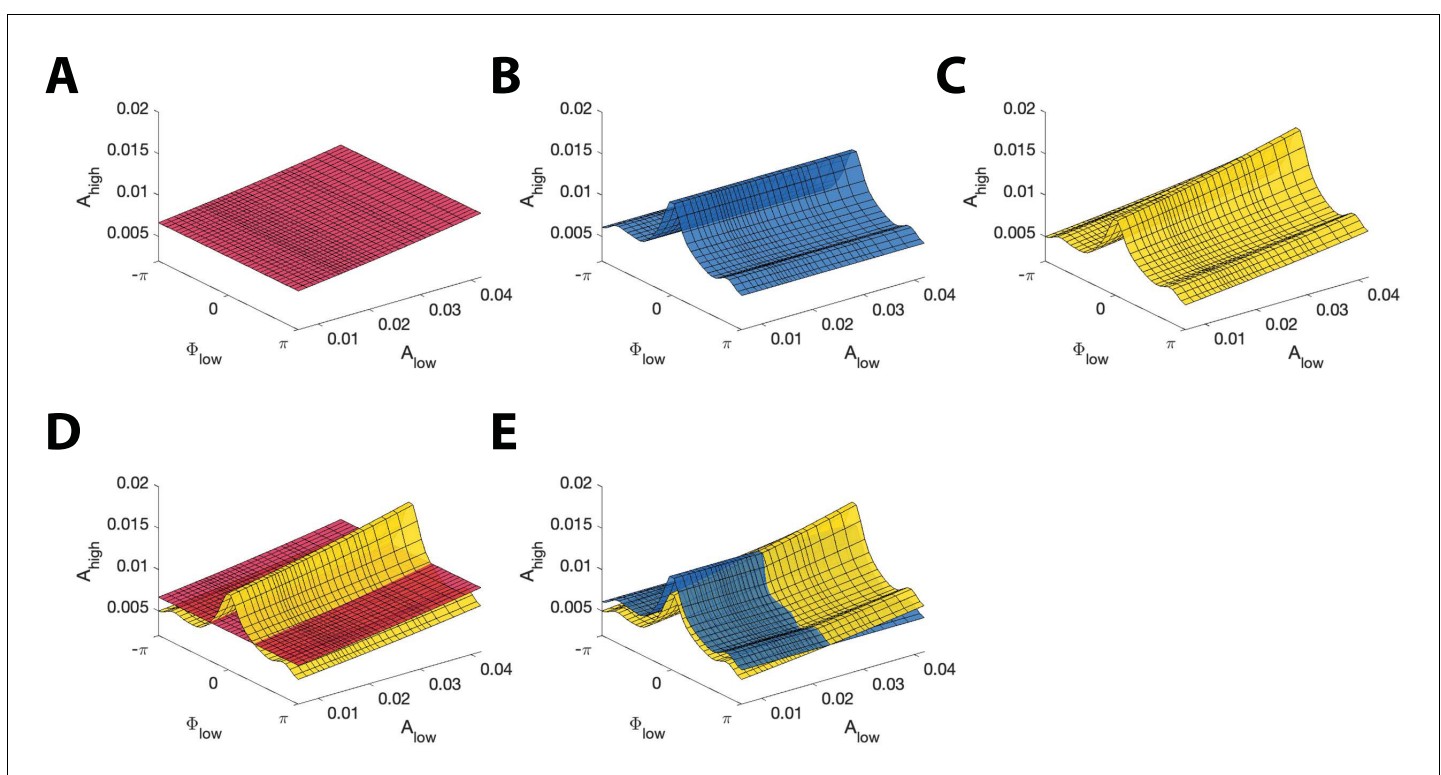

**Figure 3.** Example model surfaces used to determine $\mathbf{R}_{\mathrm{PAC}}$ and $\mathbf{R}_{\mathrm{AAC}}$. (A,B,C) Three example surfaces (A) $S_{A_{\mathrm{low}}}$, (B) $S_{\phi_{\mathrm{low}}}$, and (C) $S_{A_{\mathrm{low}},\phi_{\mathrm{low}}}$ in the three-dimensional space ($A_{\mathrm{low}}$, $\phi_{\mathrm{low}}$, $A_{\mathrm{high}}$). (D) The maximal distance between the surfaces $S_{A_{\mathrm{low}}}$ (red) and $S_{A_{\mathrm{low}},\phi_{\mathrm{low}}}$ (yellow) is used to compute $\mathbf{R}_{\mathrm{PAC}}$. (E) The maximal distance between the surfaces $S_{\phi_{\mathrm{low}}}$ (blue) and $S_{A_{\mathrm{low}},\phi_{\mathrm{low}}}$ (yellow) is used to compute $\mathbf{R}_{\mathrm{AAC}}$.

DOI: https://doi.org/10.7554/eLife.44287.004

$A_{\text{high}}$) space. The surface $S_{\phi_{\text{low}}}$ has the same structure as the curve $C_{\phi_{\text{low}}}$ in the ($\phi_{\text{low}}$, $A_{\text{high}}$) space, and remains constant along the dimension $A_{\text{low}}$ (**Figure 3B**).

The statistic $\mathbf{R}_{\text{PAC}}$ measures the effect of low frequency phase on high frequency amplitude, while accounting for fluctuations in the low frequency amplitude. To compute this statistic, we note that the model in **Equation 3** measures the combined effect of $A_{\text{low}}$ and $\phi_{\text{low}}$ on $A_{\text{high}}$, while the model in **Equation 2** measures only the effect of $A_{\text{low}}$ on $A_{\text{high}}$. Hence, to isolate the effect of $\phi_{\text{low}}$ on $A_{\text{high}}$, while accounting for $A_{\text{low}}$, we compare the difference in fits between the models in **Equations 2 and 3**. We fit the mean response functions of the models in **Equations 2 and 3**, and calculate $\mathbf{R}_{\text{PAC}}$ as the maximum absolute fractional difference between the resulting surfaces $S_{A_{\text{low}},\phi_{\text{low}}}$ and $S_{A_{\text{low}}}$ (**Figure 3D**):

$$\mathbf{R}_{\text{PAC}} = \max\left[\text{abs}\left[1 - S_{A_{\text{low}}}/S_{A_{\text{low}},\phi_{\text{low}}}\right]\right], \tag{4}$$

That is we measure the largest distance between the $A_{\text{low}}$ and the $A_{\text{low}},\phi_{\text{low}}$ models. We expect fluctuations in $S_{A_{\text{low}},\phi_{\text{low}}}$ not present in $S_{A_{\text{low}}}$ to be the result of $\phi_{\text{low}}$, that is PAC. In the absence of PAC, we expect the surfaces $S_{A_{\text{low}},\phi_{\text{low}}}$ and $S_{A_{\text{low}}}$ to be very close, resulting in a small value of $\mathbf{R}_{\text{PAC}}$. However, in the presence of PAC, we expect $S_{A_{\text{low}},\phi_{\text{low}}}$ to deviate from $S_{A_{\text{low}}}$, resulting in a large value of $\mathbf{R}_{\text{PAC}}$. We note that this measure, unlike $R_2$ metrics for linear regression, is not meant to measure the goodness-of-fit of these models to the data, but rather the differences in fits between the two models. We also note that $\mathbf{R}_{\text{PAC}}$ is an unbounded measure, as it equals the maximum absolute fractional difference between distributions, which may exceed 1.

To compute the statistic $\mathbf{R}_{\text{AAC}}$, which measures the effect of low frequency amplitude on high frequency amplitude while accounting for fluctuations in the low frequency phase, we compare the difference in fits of the model in **Equation 3** from the model in **Equation 1**. We note that the model in **Equation 3** predicts $A_{\text{high}}$ as a function of $A_{\text{low}}$ and $\phi_{\text{low}}$, while the model in **Equation 1** predicts $A_{\text{high}}$ as a function of $\phi_{\text{low}}$ only. Therefore we expect a difference in fits between the models in **Equations 1 and 3** results from the effects of $A_{\text{low}}$ on $A_{\text{high}}$. We fit the mean response functions of the models in **Equations 1 and 3** in the three-dimensional ($\phi_{\text{low}}$, $A_{\text{low}}$, $A_{\text{high}}$) space, and calculate $\mathbf{R}_{\text{AAC}}$ as the maximum absolute fractional difference between the resulting surfaces $S_{A_{\text{low}},\phi_{\text{low}}}$ and $S_{\phi_{\text{low}}}$ (**Figure 3E**):

$$\mathbf{R}_{\text{AAC}} = \max\left[\text{abs}\left[1 - S_{\phi_{\text{low}}}/S_{A_{\text{low}},\phi_{\text{low}}}\right]\right]. \tag{5}$$

That is we measure the distance between the $\phi_{\text{low}}$ and the $A_{\text{low}},\phi_{\text{low}}$ models. We expect fluctuations in $S_{A_{\text{low}},\phi_{\text{low}}}$ not present in $S_{\phi_{\text{low}}}$ to be the result of $A_{\text{low}}$, that is AAC. In the absence of AAC, we expect the surfaces $S_{A_{\text{low}},\phi_{\text{low}}}$ and $S_{\phi_{\text{low}}}$ to be very close, resulting in a small value for $\mathbf{R}_{\text{AAC}}$. Alternatively, in the presence of AAC, we expect $S_{A_{\text{low}},\phi_{\text{low}}}$ to deviate from $S_{\phi_{\text{low}}}$, resulting in a large value of $\mathbf{R}_{\text{AAC}}$.

## Estimating 95% confidence intervals for $\mathbf{R}_{\text{PAC}}$ and $\mathbf{R}_{\text{AAC}}$

We compute 95% confidence intervals for $\mathbf{R}_{\text{PAC}}$ and $\mathbf{R}_{\text{AAC}}$ via a parametric bootstrap method (**Kramer and Eden, 2013**). Given a vector of estimated coefficients $\beta_x$ for $x = \{A_{\text{low}};\ \phi_{\text{low}};\ \text{or}\ A_{\text{low}},\phi_{\text{low}}\}$, we use its estimated covariance and estimated mean to generate 10,000 normally distributed coefficient sample vectors $\beta_x^j$, $j \in \{0,\ldots,10000\}$. For each $\beta_x^j$, we then compute the high frequency amplitude values from the $A_{\text{low}}$, $\phi_{\text{low}}$, or $A_{\text{low}},\phi_{\text{low}}$ model, $S_x^j$. Finally, we compute the statistics $\mathbf{R}_{\text{PAC}}^j$ and $\mathbf{R}_{\text{AAC}}^j$ for each $j$ as,

$$\mathbf{R}_{\text{PAC}}^j = \max\left[\text{abs}\left[1 - S_{A_{\text{low}}}^j/S_{A_{\text{low}},\phi_{\text{low}}}^j\right]\right], \tag{6}$$

$$\mathbf{R}_{\text{AAC}}^j = \max\left[\text{abs}\left[1 - S_{\phi_{\text{low}}}^j/S_{A_{\text{low}},\phi_{\text{low}}}^j\right]\right]. \tag{7}$$

The 95% confidence intervals for the statistics are the values of $\mathbf{R}_{\text{PAC}}^j$ and $\mathbf{R}_{\text{AAC}}^j$ at the 0.025 and 0.975 quantiles (**Kramer and Eden, 2013**).

## Assessing significance of AAC and PAC with bootstrap p-values

To assess whether evidence exists for significant PAC or AAC, we implement a bootstrap procedure to compute p-values as follows. Given two signals $V_{\text{low}}$ and $V_{\text{high}}$, and the resulting estimated statistics $\mathbf{R}_{\text{PAC}}$ and $\mathbf{R}_{\text{AAC}}$ we apply the Amplitude Adjusted Fourier Transform (AAFT) algorithm (*Theiler et al., 1992*) on $V_{\text{high}}$ to generate a surrogate signal $V_{\text{high}}^{i}$. In the AAFT algorithm, we first reorder the values of $V_{\text{high}}$ by creating a random Gaussian signal $W$ and ordering the values of $V_{\text{high}}$ to match $W$. For example, if the highest value of $W$ occurs at index $j$, then the highest value of $V_{\text{high}}$ will be reordered to occur at index $j$. Next, we apply the Fourier Transform (FT) to the reordered $V_{\text{high}}$ and randomize the phase of the frequency domain signal. This signal is then inverse Fourier transformed and rescaled to have the same amplitude distribution as the original signal $V_{\text{high}}$. In this way, the algorithm produces a permutation $V_{\text{high}}^{i}$ of $V_{\text{high}}$ such that the power spectrum and amplitude distribution of the original signal are preserved.

We create 1000 such surrogate signals $V_{\text{high}}^{i}$, and calculate $\mathbf{R}_{\text{PAC}}^{i}$ and $\mathbf{R}_{\text{AAC}}^{i}$ between $V_{\text{low}}$ and each $V_{\text{high}}^{i}$. We define the p-values $p_{\text{PAC}}$ and $p_{\text{AAC}}$ as the proportion of values in $\{\mathbf{R}_{\text{PAC}}^{i}\}_{i=1}^{1000}$ and $\{\mathbf{R}_{\text{AAC}}^{i}\}_{i=1}^{1000}$ greater than the estimated statistics $\mathbf{R}_{\text{PAC}}$ and $\mathbf{R}_{\text{AAC}}$, respectively. If the proportion is zero, we set $p = 0.0005$.

We calculate p-values for the modulation index in the same way. The modulation index calculates the distribution of high frequency amplitudes versus low frequency phases and measures the distance from this distribution to a uniform distribution of amplitudes. Given the signals $V_{\text{low}}$ and $V_{\text{high}}$, and the resulting modulation index **MI** between them, we calculate the modulation index between $V_{\text{low}}$ and 1000 surrogate permutations of $V_{\text{high}}$ using the AAFT algorithm. We set $p_{\text{MI}}$ to be the proportion of these resulting values greater than the **MI** value estimated from the original signals.

## Synthetic time series with PAC

We construct synthetic time series to examine the performance of the proposed method as follows. First, we simulate 20 s of pink noise data such that the power spectrum scales as $1/f$. We then filter these data into low (4–7 Hz) and high (100–140 Hz) frequency bands, as described in Materials and methods: *Estimation of the phase and amplitude envelope*, creating signals $V_{\text{low}}$ and $V_{\text{high}}$. Next, we couple the amplitude of the high frequency signal to the phase of the low frequency signal. To do so, we first locate the peaks of $V_{\text{low}}$ and determine the times $t_k$, $k = \{1, 2, 3, \ldots, K\}$, of the $K$ relative extrema. We note that these times correspond approximately to $\phi_{\text{low}} = 0$. We then create a smooth modulation signal M which consists of a 42 ms Hanning window of height $1 + I_{PAC}$ centered at each $t_k$, and a value of 1 at all other times (*Figure 4A*). The intensity parameter $I_{PAC}$ in the modulation signal corresponds to the strength of PAC. $I_{PAC} = 0.0$ corresponds to no PAC, while $I_{PAC} = 1.0$ results in a 100% increase in the high frequency amplitude at each $t_k$, creating strong PAC. We create a new signal $V_{\text{high}}'$ with the same phase as $V_{\text{high}}$, but with amplitude dependent on the phase of $V_{\text{low}}$ by setting,

$$V_{\text{high}}' = \mathbf{M} V_{\text{high}} .$$

We create the final voltage trace $V$ as

$$V = V_{\text{low}} + V_{\text{high}}' + c\, V_{\text{pink}} ,$$

where $V_{\text{pink}}$ is a new instance of pink noise multiplied by a small constant $c = 0.01$. In the signal $V$, brief increases of the high frequency activity occur at a specific phase (0 radians) of the low frequency signal (*Figure 4B*).

## Synthetic time series with AAC

To generate synthetic time series with dependence on the low frequency amplitude, we follow the procedure in the preceding section to generate $V_{\text{low}}$, $V_{\text{high}}$, and $A_{\text{low}}$. We then induce amplitude-amplitude coupling between the low and high frequency components by creating a new signal $V_{\text{high}}^{*}$ such that

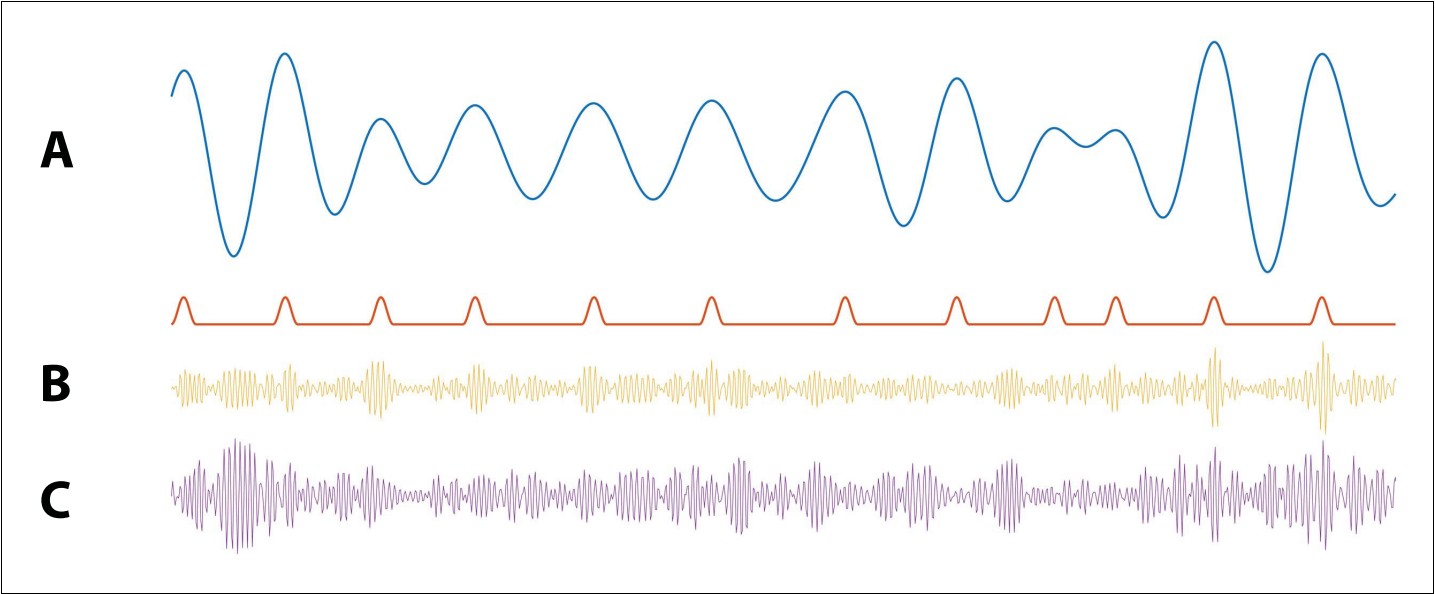

**Figure 4.** Illustration of synthetic time series with PAC and AAC. (**A**) Example simulation of $V_{\text{low}}$ (blue) and modulation signal M (red). When the phase of $V_{\text{low}}$ is near 0 radians, M increases. (**B**) Example simulation of PAC. When the phase of $V_{\text{low}}$ is approximately 0 radians, the high frequency amplitude (yellow) increases. (**C**) Example simulations of AAC. When the amplitude of $V_{\text{low}}$ is large, so is the amplitude of the high frequency signal (purple).
DOI: https://doi.org/10.7554/eLife.44287.005

$$V_{\text{high}}^* = V_{\text{high}}\left(1 + I_{\text{AAC}}\frac{A_{\text{low}}}{\max\left(A_{\text{low}}\right)}\right),$$

where $I_{\text{AAC}}$ is the intensity parameter corresponding to the strength of amplitude-amplitude coupling. We define the final voltage trace $V$ as

$$V = V_{\text{low}} + V_{\text{high}}^* + c\, V_{\text{pink}}\,,$$

where $V_{\text{pink}}$ is a new instance of pink noise multiplied by a small constant $c = 0.01$ (**Figure 4C**).

## Human subject data

A patient (male, age 32 years) with medically intractable focal epilepsy underwent clinically indicated intracranial cortical recordings for epilepsy monitoring. In addition to clinical electrode implantation, the patient was also implanted with a $10 \times 10$ ($4\,\text{mm} \times 4\,\text{mm}$) NeuroPort microelectrode array (MEA; Blackrock Microsystems, Utah) in a neocortical area expected to be resected with high probability in the temporal gyrus. The MEA consists of 96 platinum-tipped silicon probes, with a length of 1.5 mm, corresponding to neocortical layer III as confirmed by histology after resection. Signals from the MEA were acquired continuously at 30 kHz per channel. Seizure onset times were determined by an experienced encephalographer (S.S.C.) through inspection of the macroelectrode recordings, referral to the clinical report, and clinical manifestations recorded on video. For a detailed clinical summary, see patient P2 of *Wagner et al. (2015)*. For these data, we analyze the 100–140 Hz and 4–7 Hz frequency bands to illustrate the proposed method; a more rigorous study of CFC in these data may require a more principled choice of high frequency band. All patients were enrolled after informed consent and consent to publish was obtained, and approval was granted by local Institutional Review Boards at Massachusetts General Hospital and Brigham Women's Hospitals (Partners Human Research Committee), and at Boston University according to National Institutes of Health guidelines.

### Code availability
The code to perform this analysis is available for reuse and further development at https://github.com/Eden-Kramer-Lab/GLM-CFC (*Nadalin and Kramer, 2019*; copy archived at https://github.com/elifesciences-publications/GLM-CFC).

## Results

We first examine the performance of the CFC measure through simulation examples. In doing so, we show that the statistics $\mathbf{R}_{\mathrm{PAC}}$ and $\mathbf{R}_{\mathrm{AAC}}$ accurately detect different types of cross-frequency coupling, increase with the intensity of coupling, and detect weak PAC coupled to the low frequency amplitude. We show that the proposed method is less sensitive to changes in low frequency power, and outperforms an existing PAC measure that lacks dependence on the low frequency amplitude. We conclude with example applications to human and rodent in vivo recordings, and show how to extend the modeling framework to include a new covariate.

### The absence of CFC produces no significant detections of coupling

We first consider simulated signals without CFC. To create these signals, we follow the procedure in Materials and methods: *Synthetic Time Series with PAC* with the modulation intensity set to zero ($I_{PAC} = 0$). In the resulting signals, $A_{\mathrm{high}}$ is approximately constant and does not depend on $\phi_{\mathrm{low}}$ or $A_{\mathrm{low}}$ (*Figure 5A*). We estimate the $\phi_{\mathrm{low}}$ model, the $A_{low}$ model, and the $A_{\mathrm{low}}, \phi_{\mathrm{low}}$ model from these data; we show example fits of the model surfaces in *Figure 5B*. We observe that the models exhibit small modulations in the estimated high frequency amplitude envelope as a function of the low frequency phase and amplitude.

To assess the distribution of significant $\mathbf{R}$ values in the case of no cross-frequency coupling, we simulate 1000 instances of the pink noise signals (each of 20 s) and apply the $\mathbf{R}$ measures to each instance, plotting significant $\mathbf{R}$ values in *Figure 5C*. We find that for all 1000 instances, $p_{\mathrm{PAC}}$ and $p_{\mathrm{AAC}}$ are less than 0.05 in only 0.6% and 0.2% of the simulations, respectively, indicating no significant evidence of PAC or AAC, as expected.

We also applied these simulated signals to assess the performance of two standard model comparison procedures for GLMs. Simulating 1000 instances of pink noise signals (each of 20 s) with no induced PAC or AAC, we performed a chi-squared test for nested models (*Kramer and Eden, 2016*) between models $A_{\mathrm{low}}$ and $A_{\mathrm{low}}, \phi_{\mathrm{low}}$, and detected significant PAC ($p < 0.05$) in 59.7% of simulations. Similarly, performing a chi-squared test for nested models between models $\phi_{\mathrm{low}}$ and $A_{\mathrm{low}}, \phi_{\mathrm{low}}$, we detected significant AAC ($p < 0.05$) in 41.5% of simulations. Using an AIC-based model comparison, we found a decrease in AIC from the $A_{\mathrm{low}}$ model to the $A_{\mathrm{low}}, \phi_{\mathrm{low}}$ model (consistent with significant PAC) in 98.6% of simulations, and a decrease in AIC from the $\phi_{\mathrm{low}}$ model to the $A_{\mathrm{low}}, \phi_{\mathrm{low}}$ model (consistent with significant AAC) in 87.2% of simulations. By contrast, we rarely detect significant PAC (<0.6% of simulations) or AAC (<0.2% of simulations) in the pink noise signals using the two statistics $\mathbf{R}_{\mathrm{PAC}}$ and $\mathbf{R}_{\mathrm{AAC}}$ implemented here. We conclude that, in this modeling regime, two deviance-based model comparison procedures for GLMs are less robust measures of significant PAC and AAC.

### The proposed method accurately detects PAC

We next consider signals that possess phase-amplitude coupling, but lack amplitude-amplitude coupling. To do so, we simulate a 20 s signal with $A_{\mathrm{high}}$ modulated by $\phi_{\mathrm{low}}$ (*Figure 5D*); more specifically, $A_{\mathrm{high}}$ increases when $\phi_{\mathrm{low}}$ is near 0 radians (see Materials and methods, $I_{\mathrm{PAC}} = 1$). We then estimate the $\phi_{\mathrm{low}}$ model, the $A_{\mathrm{low}}$ model, and the $A_{\mathrm{low}}, \phi_{\mathrm{low}}$ model from these data; we show example fits in *Figure 5E*. We find that in the $\phi_{\mathrm{low}}$ model $A_{\mathrm{high}}$ is higher when $\phi_{\mathrm{low}}$ is close to 0 radians, and the $A_{\mathrm{low}}, \phi_{\mathrm{low}}$ model follows this trend. We note that, because the data do not depend on the low frequency amplitude ($A_{\mathrm{low}}$), the $\phi_{\mathrm{low}}$ and $A_{\mathrm{low}}, \phi_{\mathrm{low}}$ models have very similar shapes in the ($\phi_{\mathrm{low}}, A_{\mathrm{low}}, A_{\mathrm{high}}$) space, and the $A_{\mathrm{low}}$ model is nearly flat.

Simulating 1000 instances of these 20 s signals with induced phase-amplitude coupling, we find $p_{\mathrm{AAC}} < 0.05$ for only 0.6% of the simulations, while $p_{\mathrm{PAC}} < 0.05$ for 96.5% of the simulations. We find that the significant values of $\mathbf{R}_{\mathrm{PAC}}$ lie well above 0 (*Figure 5F*), and that as the intensity of the simulated phase-amplitude coupling increases, so does the statistic $\mathbf{R}_{\mathrm{PAC}}$ (*Figure 5G*). We conclude that

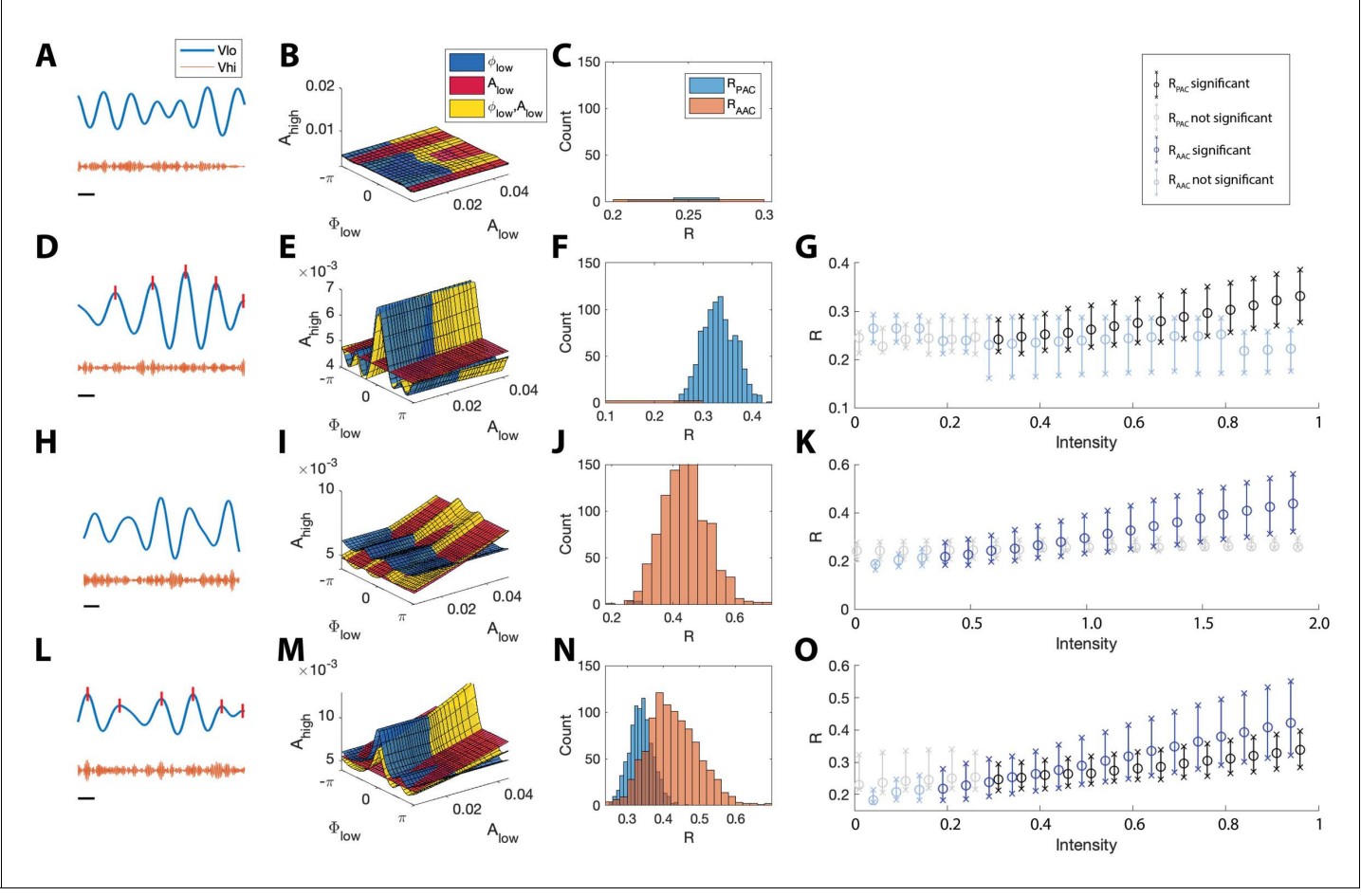

**Figure 5.** The statistical modeling framework successfully detects different types of cross-frequency coupling. (**A–C**) Simulations with no CFC. (**A**) When no CFC occurs, the low frequency signal (blue) and high frequency signal (orange) evolve independently. (**B**) The surfaces $S_{A_{low}}$, $S_{\phi_{low}}$, and $S_{A_{low},\phi_{low}}$ suggest no dependence of $A_{high}$ on $\phi_{low}$ or $A_{low}$. (**C**) Significant ($p<0.05$) values of $\mathbf{R}_{PAC}$ and $\mathbf{R}_{AAC}$ from 1000 simulations. Very few significant values for the statistics R are detected. (**D–G**) Simulations with PAC only. (**D**) When the phase of the low frequency signal is near 0 radians (red tick marks), the amplitude of the high frequency signal increases. (**E**) The surfaces $S_{A_{low}}$, $S_{\phi_{low}}$, and $S_{A_{low},\phi_{low}}$ suggest dependence of $A_{high}$ on $\phi_{low}$. (**F**) In 1000 simulations, significant values of $\mathbf{R}_{PAC}$ frequently appear, while significant values of $\mathbf{R}_{AAC}$ rarely appear. (**G**) As the intensity of PAC increases, so do the significant values of $\mathbf{R}_{PAC}$ (black), while any significant values of $\mathbf{R}_{AAC}$ remain small. (**H–K**) Simulations with AAC only. (**H**) The amplitudes of the high frequency signal and low frequency signal are positively correlated. (**I**) The surfaces $S_{A_{low}}$, $S_{\phi_{low}}$, and $S_{A_{low},\phi_{low}}$ suggest dependence of $A_{high}$ on $A_{low}$. (**J**) In 1000 simulations, significant values of $\mathbf{R}_{AAC}$ frequently appear. (**K**) As the intensity of AAC increases, so do the significant values of $\mathbf{R}_{AAC}$ (blue), while any significant values of $\mathbf{R}_{PAC}$ remain small. (**L–O**) Simulations with PAC and AAC. (**L**) The amplitude of the high frequency signal increases when the phase of the low frequency signal is near 0 radians and the amplitude of the low frequency signal is large. (**M**) The surfaces $S_{A_{low}}$, $S_{\phi_{low}}$, and $S_{A_{low},\phi_{low}}$ suggest dependence of $A_{high}$ on $\phi_{low}$ and $A_{low}$. (**N**) In 1000 simulations, significant values of $\mathbf{R}_{PAC}$ and $\mathbf{R}_{AAC}$ frequently appear. (**O**) As the intensity of PAC and AAC increase, so do the significant values of $\mathbf{R}_{PAC}$ and $\mathbf{R}_{AAC}$. In (G,K,O), circles indicate the median, and x's the 5th and 95th quantiles.
DOI: https://doi.org/10.7554/eLife.44287.006

the proposed method accurately detects the presence of phase-amplitude coupling in these simulated data.

## The proposed method accurately detects AAC

We next consider signals with amplitude-amplitude coupling, but without phase-amplitude coupling. We simulate a 20 s signal such that $A_{high}$ is modulated by $A_{low}$ (see Materials and methods, $I_{AAC} = 1$); when $A_{low}$ is large, so is $A_{high}$ (**Figure 5H**). We then estimate the $\phi_{low}$ model, the $A_{low}$ model, and the $A_{low}, \phi_{low}$ model (example fits in **Figure 5I**). We find that the $A_{low}$ model increases along the $A_{low}$ axis, and that the $A_{low}, \phi_{low}$ model closely follows this trend, while the $\phi_{low}$ model remains mostly flat, as expected.

Simulating 1000 instances of these signals we find that $p_{\text{AAC}}<0.05$ for 97.9% of simulations, while $p_{\text{PAC}}<0.05$ for 0.3% of simulations. The significant values of $\mathbf{R}_{\text{AAC}}$ lie above 0 (*Figure 5J*), and increases in the intensity of AAC produce increases in $\mathbf{R}_{\text{AAC}}$ (*Figure 5K*). We conclude that the proposed method accurately detects the presence of amplitude-amplitude coupling.

## The proposed method accurately detects the simultaneous occurrence of PAC and AAC

We now consider signals that possess both phase-amplitude coupling and amplitude-amplitude coupling. To do so, we simulate time series data with both AAC and PAC (*Figure 5L*). In this case, $A_{\text{high}}$ increases when $\phi_{low}$ is near 0 radians and when $A_{\text{low}}$ is large (see Materials and methods, $I_{\text{PAC}} = 1$ and $I_{\text{AAC}} = 1$). We then estimate the $\phi_{\text{low}}$ model, the $A_{low}$ model, and the $A_{\text{low}}, \phi_{\text{low}}$ model from the data and visualize the results (*Figure 5M*). We find that the $\phi_{\text{low}}$ model increases near $\phi_{\text{low}} = 0$, and that the $A_{\text{low}}$ model increases linearly with $A_{\text{low}}$. The $A_{\text{low}}, \phi_{\text{low}}$ model exhibits both of these behaviors, increasing at $\phi_{\text{low}} = 0$ and as $A_{\text{low}}$ increases.

Simulating 1000 instances of signals with both AAC and PAC present, we find that $p_{\text{AAC}}<0.05$ in 96.7% of simulations and $p_{PAC}<0.05$ in 98.1% of simulations. The distributions of significant $\mathbf{R}_{\text{PAC}}$ and $\mathbf{R}_{\text{AAC}}$ values lie above 0, consistent with the presence of both PAC and AAC (*Figure 5N*), and as the intensity of PAC and AAC increases, so do the values of $\mathbf{R}_{\text{PAC}}$ and $\mathbf{R}_{\text{AAC}}$ (*Figure 5O*). We conclude that the model successfully detects the concurrent presence of PAC and AAC.

## $\mathbf{R}_{\text{PAC}}$ and modulation index are both sensitive to weak modulations

To investigate the ability of the proposed method and the modulation index to detect weak coupling between the low frequency phase and high frequency amplitude, we perform the following simulations. For each intensity value $I_{\text{PAC}}$ between 0 and 0.5 (in steps of 0.025), we simulate 1000 signals (see Materials and methods) and compute $\mathbf{R}_{\text{PAC}}$ and a measure of PAC in common use: the modulation index **MI** (*Tort et al., 2010*) (*Figure 6*). We find that both **MI** and $\mathbf{R}_{\text{PAC}}$, while small, increase with $I_{PAC}$; in this way, both measures are sensitive to small values of $I_{PAC}$. However, we note that $\mathbf{R}_{\text{PAC}}$ is not significant for very small intensity values ($I_{\text{PAC}} \leq 0.3$), while **MI** is significant at these small intensities. Significant $\mathbf{R}_{\text{PAC}}$ appears when the **MI** exceeds $0.7 \times 10^{-3}$, a value below the range of **MI** values detected in many existing studies (*Tort et al., 2008*; *Zhong et al., 2017*; *Jackson et al., 2019*; *Axmacher et al., 2010*; *Tort et al., 2018*). We conclude that, while the modulation index may be more sensitive than $\mathbf{R}_{\text{PAC}}$ to very weak phase-amplitude coupling, $\mathbf{R}_{\text{PAC}}$ can detect phase-amplitude coupling at **MI** values consistent with those observed in the literature.

## The proposed method is less affected by fluctuations in low-frequency amplitude and AAC

Increases in low frequency power can increase measures of phase-amplitude coupling, although the underlying PAC remains unchanged (*Aru et al., 2015*; *Cole and Voytek, 2017*). Characterizing the impact of this confounding effect is important both to understand measure performance and to produce accurate interpretations of analyzed data. To examine this phenomenon, we perform the following simulation. First, we simulate a signal $V$ with fixed PAC (intensity $I_{PAC} = 1$, see Materials and methods). Second, we filter $V$ into its low and high frequency components $V_{\text{low}}$ and $V_{\text{high}}$, respectively. Then, we create a new signal $V^*$ as follows:

$$V^* = 2\,V_{\text{low}} + V_{\text{high}} + V_{\text{noise}}, \tag{8}$$

where $V_{\text{noise}}$ is a pink noise term (see Materials and methods). We note that we only alter the low frequency component of $V$ and do not alter the PAC. To analyze the PAC in this new signal we compute $\mathbf{R}_{\text{PAC}}$ and **MI**.

We show in *Figure 7* population results (1000 realizations each of the simulated signals $V$ and $V^*$) for the R and **MI** values. We observe that increases in the amplitude of $V_{\text{low}}$ produce increases in **MI** and $\mathbf{R}_{\text{PAC}}$. However, this increase is more dramatic for **MI** than for $\mathbf{R}_{\text{PAC}}$; we note that the distributions of $\mathbf{R}_{PAC}$ almost completely overlap (*Figure 7A*), while the distribution of **MI** shifts to larger values when the amplitude of $V_{\text{low}}$ increases (*Figure 7B*). We conclude that the statistic $\mathbf{R}_{\text{PAC}}$ — which includes the low frequency amplitude as a predictor in the GLM — is more robust to increases in low frequency power than a method that only includes the low frequency phase.

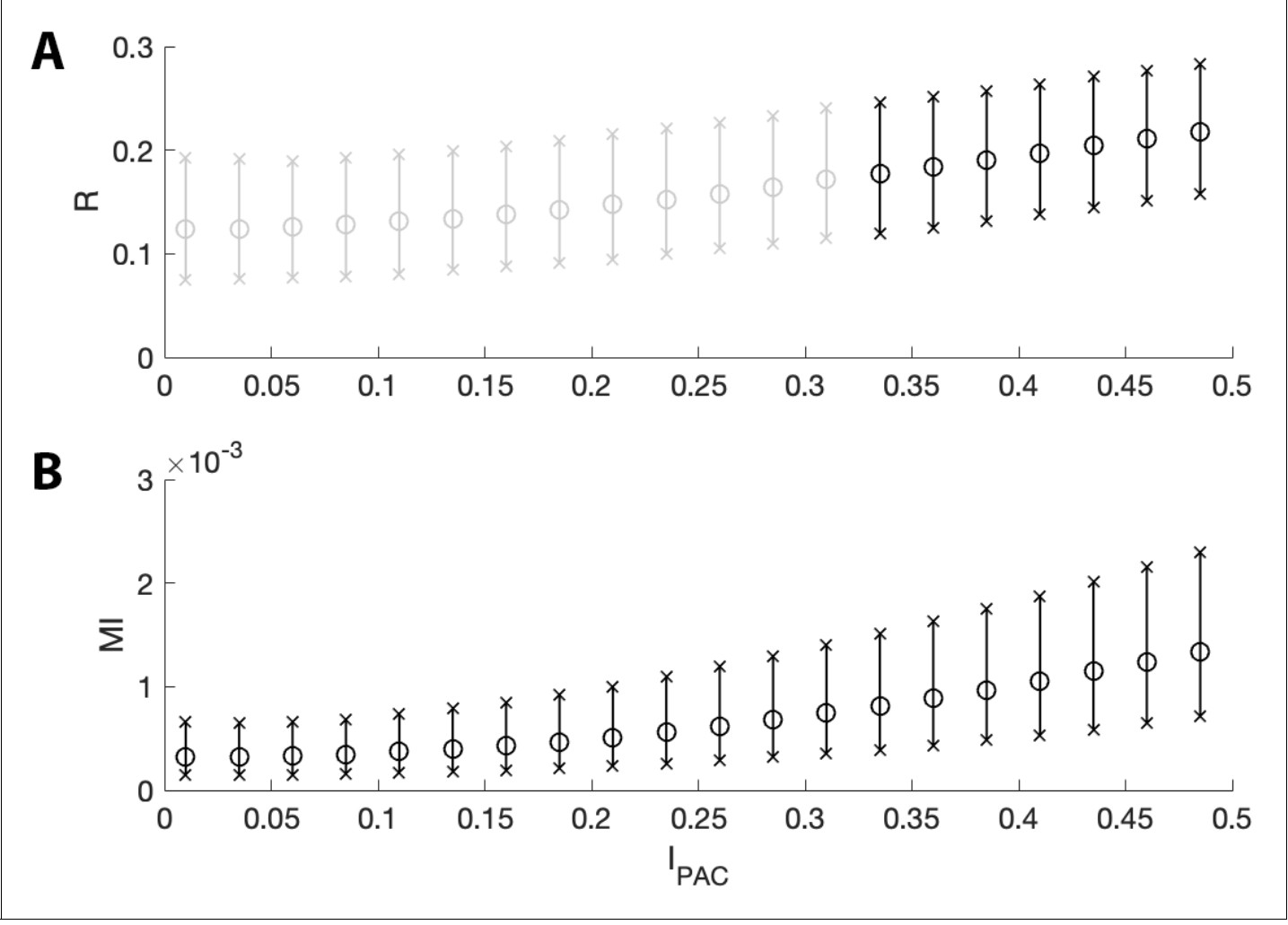

**Figure 6.** The two measures of PAC increase with intensities near zero. The mean (circles) and 5[th] to 95[th] quantiles (x's) of (A) $\mathbf{R}_{PAC}$ and (B) MI for intensity values between 0 and 0.5. Black bars indicate $p_{PAC}$ or $p_{MI}$ is below 0.05 for ≥95% of simulations; gray bars indicate $p_{PAC}$ is not below 0.05 for ≥95% of simulations. While both measures increase with intensity, MI detects more instances of significant PAC than does $\mathbf{R}_{PAC}$ for very small values of $I_{PAC}$.

DOI: https://doi.org/10.7554/eLife.44287.007

We also investigate the effect of increases in amplitude-amplitude coupling (AAC) on the two measures of PAC. As before, we simulate a signal $V$ with fixed PAC (intensity $I_{PAC} = 1$) and no AAC (intensity $I_{AAC} = 0$). We then simulate a second signal $V^*$ with the same fixed PAC as $V$, and with additional AAC (intensity $I_{AAC} = 10$). We simulate 1000 realizations of $V$ and $V^*$ and compute the corresponding $\mathbf{R}_{PAC}$ and MI values. We observe that the increase in AAC produces a small increase in the distribution of $\mathbf{R}_{PAC}$ values (*Figure 7C*), but a large increase in the distribution of MI values (*Figure 7D*). We conclude that the statistic $\mathbf{R}_{PAC}$ is more robust to increases in AAC than MI.

These simulations show that at a fixed, non-zero PAC, the modulation index increases with increased $A_{low}$ and AAC. We now consider the scenario of increased $A_{low}$ and AAC in the absence of PAC. To do so, we simulate 1000 signals of 200 s duration, with no PAC (intensity $I_{PAC} = 0$). For each signal, at time 100 s (i.e., the midpoint of the simulation) we increase the low frequency amplitude by a factor of 10 (consistent with observations from an experiment in rodent cortex, as described below), and include AAC between the low and high frequency signals (intensity $I_{AAC} = 0$ for $t < 100$ s and intensity $I_{AAC} = 2$ for $t \geq 100$ s). We find that, in the absence of PAC, $\mathbf{R}_{PAC}$ detects significant PAC ($p < 0.05$) in 0.4% of the simulated signals, while MI detects significant PAC in 34.3% of simulated signals. We conclude that in the presence of increased low frequency amplitude and

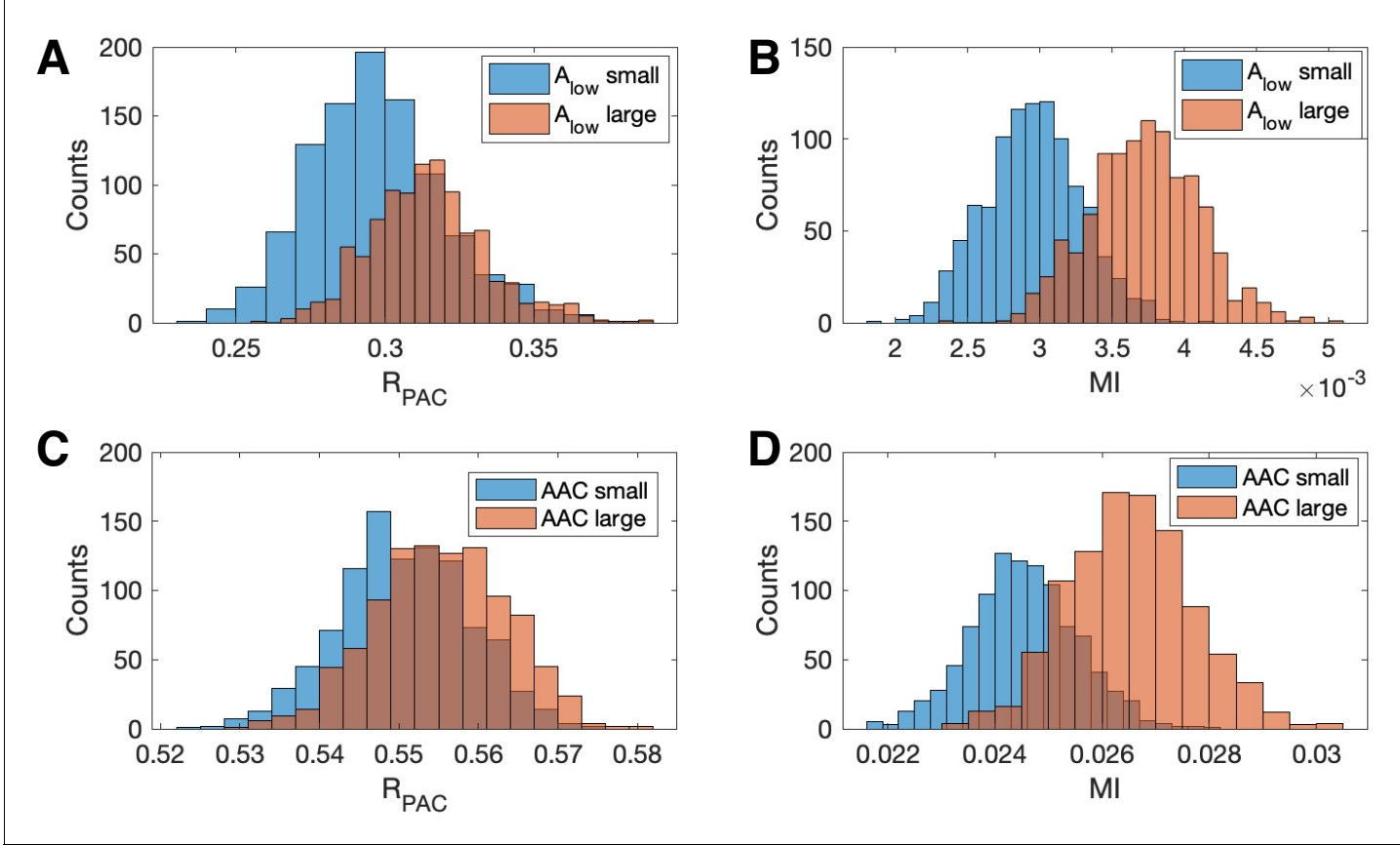

**Figure 7.** Increases in the amplitude of the low frequency signal, and the amplitude-amplitude coupling (AAC), increase the modulation index more than $\mathbf{R}_{\mathrm{PAC}}$. (A,B) Distributions of (A) $\mathbf{R}_{\mathrm{PAC}}$ and (B) MI when $A_{\mathrm{low}}$ is small (blue) and when $A_{\mathrm{low}}$ is large (red). (C,D) Distributions of (C) $\mathbf{R}_{\mathrm{PAC}}$ and (D) MI when AAC is small (blue) and when AAC is large (red).

DOI: https://doi.org/10.7554/eLife.44287.008

amplitude-amplitude coupling, **MI** may detect PAC where none exists, while $\mathbf{R}_{\mathrm{PAC}}$, which accounts for fluctuations in low frequency amplitude, does not.

## Sparse PAC is detected when coupled to the low frequency amplitude

While the modulation index has been successfully applied in many contexts (*Canolty and Knight, 2010*; *Hyafil et al., 2015b*), instances may exist where this measure is not optimal. For example, because the modulation index was not designed to account for the low frequency amplitude, it may fail to detect PAC when $A_{\mathrm{high}}$ depends not only on $\phi_{\mathrm{low}}$, but also on $A_{\mathrm{low}}$. For example, since the modulation index considers the distribution of $A_{\mathrm{high}}$ at all observed values of $\phi_{\mathrm{low}}$, it may fail to detect coupling events that occur sparsely at only a subset of appropriate $\phi_{\mathrm{low}}$ occurrences. $\mathbf{R}_{\mathrm{PAC}}$, on the other hand, may detect these sparse events if these events are coupled to $A_{\mathrm{low}}$, as $\mathbf{R}_{\mathrm{PAC}}$ accounts for fluctuations in low frequency amplitude. To illustrate this, we consider a simulation scenario in which PAC occurs sparsely in time.

We create a signal $V$ with PAC, and corresponding modulation signal **M** with intensity value $I_{\mathrm{PAC}} = 1.0$ (see Materials and methods, *Figure 8A–B*). We then modify this signal to reduce the number of PAC events in a way that depends on $A_{\mathrm{low}}$. To do so, we preserve PAC at the peaks of $V_{\mathrm{low}}$ (i. e., when $\phi_{\mathrm{low}} = 0$), but now only when these peaks are large, more specifically in the top 5% of peak values.

We define a threshold value $T$ to be the 95[th] quantile of the peak $V_{\mathrm{low}}$ values, and modify the modulation signal **M** as follows. When **M** exceeds 1 (i.e., when $\phi_{\mathrm{low}} = 0$) and the low frequency amplitude exceeds $T$ (i.e., $A_{\mathrm{low}} \geq T$), we make no change to **M**. Alternatively, when **M** exceeds one and the low frequency amplitude lies below $T$ (i.e., $A_{\mathrm{low}} < T$), we decrease **M** to 1 (*Figure 8C*). In this

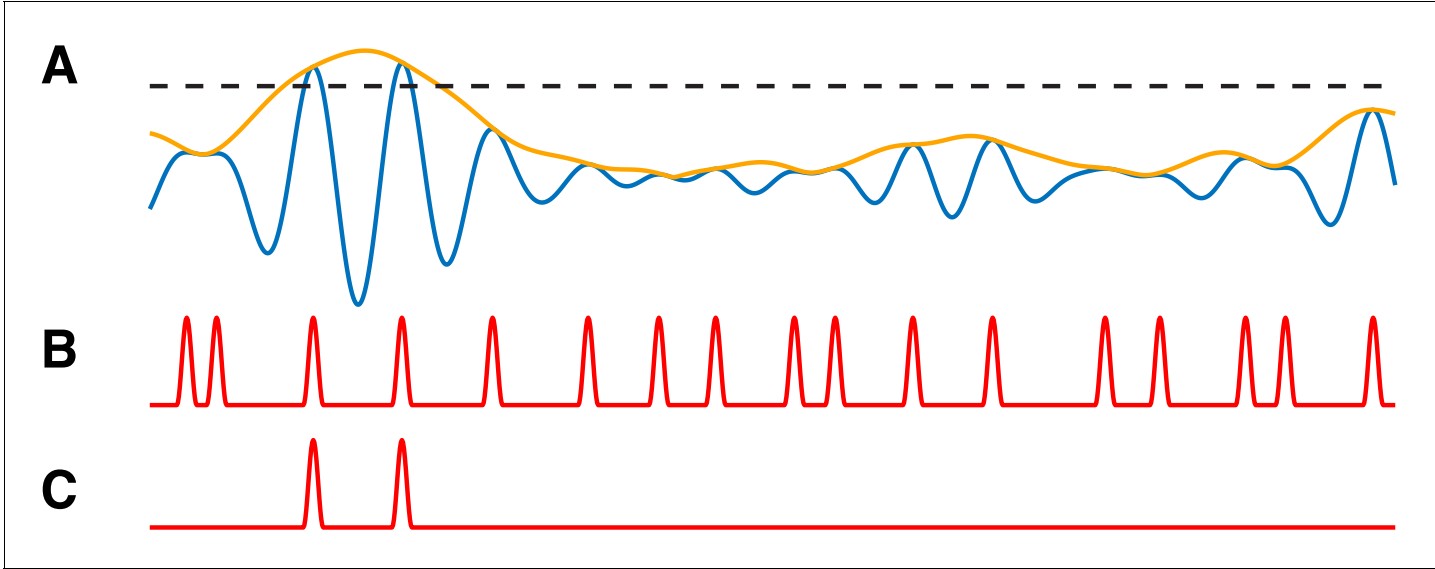

**Figure 8.** PAC events restricted to a subset of occurrences are still detectable. (**A**) The low frequency signal (blue), amplitude envelope (yellow), and threshold (black dashed). (**B–C**) The modulation signal increases (**B**) at every occurrence of $\phi_{\text{low}} = 0$, or (**C**) only when $A_{\text{low}}$ exceeds the threshold and $\phi_{\text{low}} = 0$.

DOI: https://doi.org/10.7554/eLife.44287.009

way, we create a modified modulation signal $\mathbf{M}_1$ such that in the resulting signal $V_1$, when $\phi_{\text{low}} = 0$ and $A_{\text{low}}$ is large enough, $A_{\text{high}}$ is increased; and when $\phi_{\text{low}} = 0$ and $A_{\text{low}}$ is not large enough, there is no change to $A_{\text{high}}$. This signal $V_1$ hence has fewer phase-amplitude coupling events than the number of times $\phi_{\text{low}} = 0$.

We generate 1000 realizations of the simulated signals $V_1$, and compute $\mathbf{R}_{\text{PAC}}$ and $\mathbf{MI}$. We find that while $\mathbf{MI}$ detects significant PAC in only 37% of simulations, $\mathbf{R}_{\text{PAC}}$ detects significant PAC in 72% of simulations. In this case, although the PAC occurs infrequently, these occurrences are coupled to $A_{\text{low}}$, and $\mathbf{R}_{\text{PAC}}$, which accounts for changes in $A_{\text{low}}$, successfully detects these events much more frequently. We conclude that when the PAC is dependent on $A_{\text{low}}$, $\mathbf{R}_{\text{PAC}}$ more accurately detects these sparse coupling events.

## The CFC model detects simultaneous PAC and AAC missed in an existing method

To further illustrate the utility of the proposed method, we consider another scenario in which $A_{\text{low}}$ impacts the occurrence of PAC. More specifically, we consider a case in which $A_{\text{high}}$ increases at a fixed low frequency phase for high values of $A_{\text{low}}$, and $A_{\text{high}}$ decreases at the same phase for small values of $A_{\text{low}}$. In this case, we expect that the modulation index may fail to detect the coupling because the distribution of $A_{\text{high}}$ over $\phi_{\text{low}}$ would appear uniform when averaged over all values of $A_{\text{low}}$; the dependence of $A_{\text{high}}$ on $\phi_{\text{low}}$ would only become apparent after accounting for $A_{\text{low}}$.

To implement this scenario, we consider the modulation signal $\mathbf{M}$ (see Materials and methods) with an intensity value $I_{\text{PAC}} = 1$. We consider all peaks of $A_{\text{low}}$ and set the threshold $T$ to be the 50th quantile (*Figure 9A*). We then modify the modulation signal $\mathbf{M}$ as follows. When $\mathbf{M}$ exceeds 1 (i.e., when $\phi_{\text{low}} = 0$) and the low frequency amplitude exceeds $T$ (i.e., $A_{\text{low}} \geq T$), we make no change to $\mathbf{M}$. Alternatively, when $\mathbf{M}$ exceeds one and the low frequency amplitude lies below $T$ (i.e. $A_{\text{low}} < T$), we decrease $\mathbf{M}$ to 0 (*Figure 9B*). In this way, we create a modified modulation signal $\mathbf{M}$ such that when $\phi_{\text{low}} = 0$ and $A_{\text{low}}$ is large enough, $A_{\text{high}}$ is increased; and when $\phi_{\text{low}} = 0$ and $A_{\text{low}}$ is small enough, $A_{\text{high}}$ is decreased (*Figure 9C*).

Using this method, we simulate 1000 realizations of this signal, and calculate $\mathbf{MI}$ and $\mathbf{R}_{\text{PAC}}$ for each signal (*Figure 9D*). We find that $\mathbf{R}_{\text{PAC}}$ detects significant PAC in nearly all (96%) of the simulations, while $\mathbf{MI}$ detects significant PAC in only 58% of the simulations. We conclude that, in this simulation, $\mathbf{R}_{\text{PAC}}$ more accurately detects PAC coupled to low frequency amplitude.

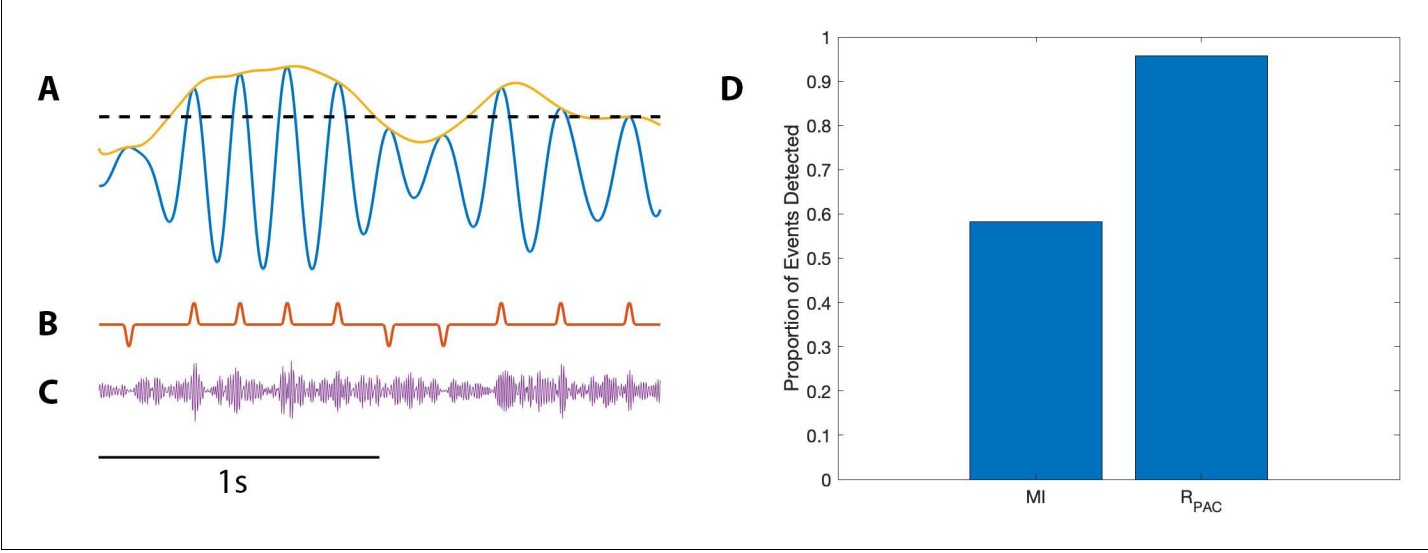

**Figure 9.** PAC with AAC is accurately detected with the proposed method, but not with the modulation index. (**A**) The low frequency signal (blue), amplitude envelope (yellow), and threshold (black dashed). (**B**) The modulation signal (red) increases when $\phi_{\mathrm{low}} = 0$ and $A_{\mathrm{low}} > T$, and deceases when $\phi_{\mathrm{low}} = 0$ and $A_{\mathrm{low}} < T$. (**C**) The modulated $A_{\mathrm{high}}$ signal (purple) increases and decreases with the modulation signal. (**D**) The proportion of significant detections (out of 1000) for **MI** and **R**$_{\mathrm{PAC}}$.

DOI: https://doi.org/10.7554/eLife.44287.010

## A simple stochastic spiking neural model illustrates the utility of the proposed method

In the previous simulations, we created synthetic data without a biophysically principled generative model. Here we consider an alternative simulation strategy with a more direct connection to neural dynamics. While many biophysically motivated models of cross-frequency coupling exist (*Sase et al., 2017*; *Chehelcheraghi et al., 2017*; *Sotero, 2016*; *Hyafil et al., 2015a*; *Lepage and Vijayan, 2015*; *Onslow et al., 2014*; *Fontolan et al., 2013*; *Malerba and Kopell, 2013*; *Jirsa and Müller, 2013*; *Spaak et al., 2012*; *Wulff et al., 2009*; *Tort et al., 2007*), we consider here a relatively simple stochastic spiking neuron model (*Aljadeff et al., 2016*). In this stochastic model, we generate a spike train ($V_{\mathrm{high}}$) in which an externally imposed signal $V_{\mathrm{low}}$ modulates the probability of spiking as a function of $A_{\mathrm{low}}$ and $\phi_{\mathrm{low}}$. We note that high frequency activity is thought to represent the aggregate spiking activity of local neural populations (*Ray and Maunsell, 2011*; *Buzsáki and Wang, 2012*; *Ray et al., 2008a*; *Jia and Kohn, 2011*); while here we simulate the activity of a single neuron, the spike train still produces temporally focal events of high frequency activity. In this framework, we allow the target phase ($\phi_{\mathrm{low}}^{*}$) modulating $A_{\mathrm{high}}$ to change as a function of $A_{\mathrm{low}}$: when $A_{\mathrm{low}}$ is large, the probability of spiking is highest near $\phi_{\mathrm{low}} = \pm\pi$, and when $A_{\mathrm{low}}$ is small, the probability of spiking is highest near $\phi_{\mathrm{low}} = 0$. More precisely, we define $\phi_{\mathrm{low}}^{*}$ as

$$\phi_{\mathrm{low}}^{*} = \pi(1 + A_{\mathrm{low}})$$

where $A_{\mathrm{low}}$ is a sinusoid oscillating between 1 and 2 with period 0.1 Hz. We define the spiking probability, $\lambda$, as

$$\lambda = \lambda_0 \exp\left[-\left(1 + \frac{s(\phi_{\mathrm{low}} - \phi_{\mathrm{low}}^{*})^2}{2\sigma^2}\right)\right],$$

where $\sigma = 0.01$, $s(\phi)$ is a triangle wave, and we choose $\lambda_0$ so that the maximum value of $\lambda$ is 2. We note that the spiking probability $\lambda$ is zero except near times when the phase of the low frequency signal ($\phi_{\mathrm{low}}$) is near $\phi_{\mathrm{low}}^{*}$. We then define $A_{\mathrm{high}}$ as:

$$A_{\mathrm{high}} = S + n,$$

where $S$ is the binary sequence generated by the stochastic spiking neuron model, and $n$ is Gaussian noise with mean zero and standard deviation 0.1. In this scenario, the distribution of $A_{\text{high}}$ over $\phi_{\text{low}}$ appears uniform when averaged over all values of $A_{\text{low}}$. We therefore expect the modulation index to remain small, despite the presence of PAC with maximal phase dependent on $A_{\text{low}}$. However, we expect that $\mathbf{R}_{\text{PAC}}$, which accounts for fluctuations in low frequency amplitude, will detect this PAC. We show an example signal from this simulation in *Figure 10A*. As expected, we find that $\mathbf{R}_{\text{PAC}}$ detects PAC ($\mathbf{R}_{\text{PAC}} = 0.172$, $p = 0.02$); we note that the $(A_{\text{low}}, \phi_{\text{low}})$ surface exhibits a single peak near $\phi_{\text{low}} = 0$ at small values of $A_{\text{low}}$, and at $\phi_{\text{low}} = \pm\pi$ at large value of $A_{\text{low}}$ (*Figure 10B*). The $(A_{\text{low}}, \phi_{\text{low}})$ surface deviates significantly from the $A_{\text{low}}$ surface, resulting in a large $\mathbf{R}_{\text{PAC}}$ value. However, the

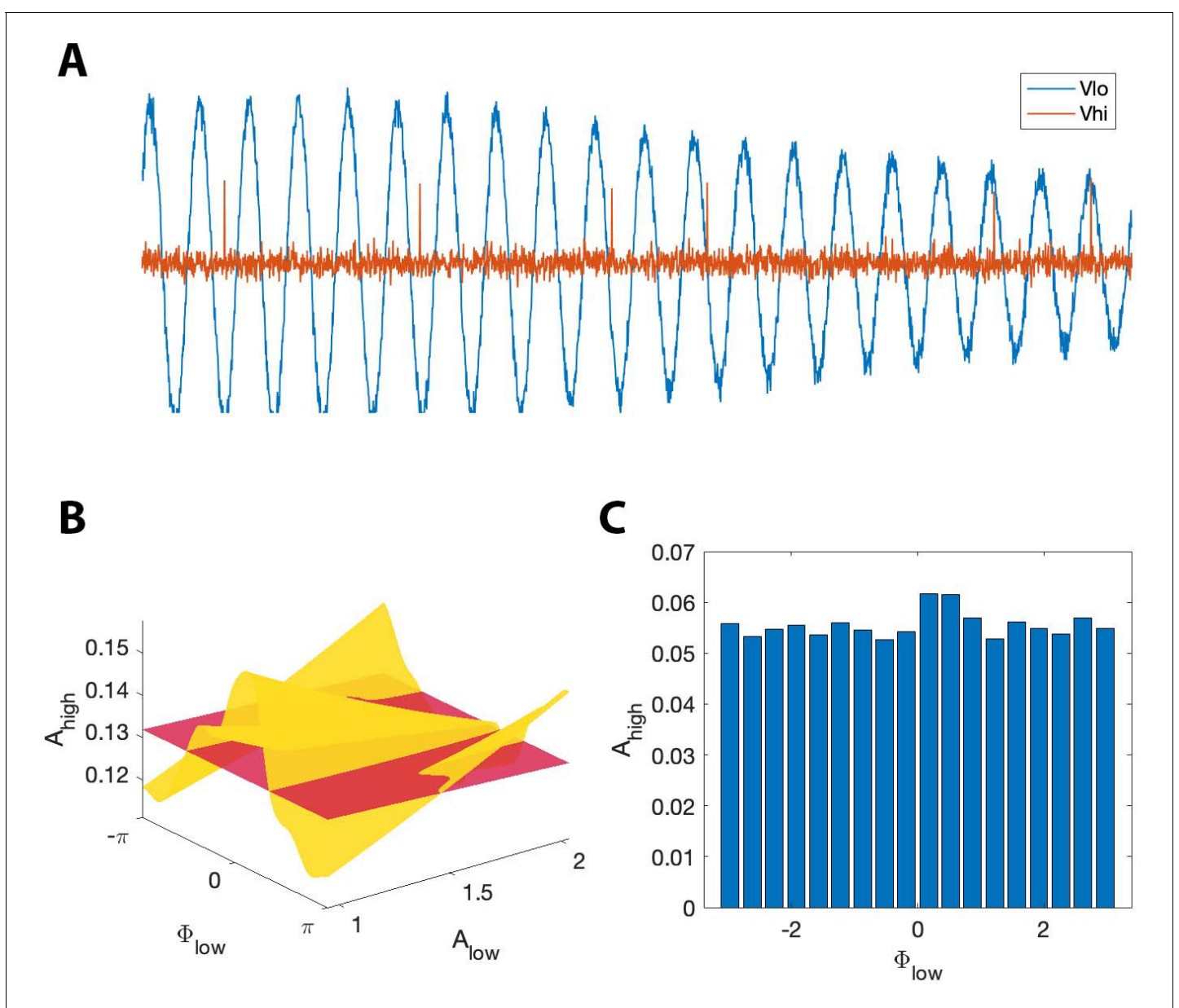

**Figure 10.** $\mathbf{R}_{\text{PAC}}$, but not MI, detects phase-amplitude coupling in a simple stochastic spiking neuron model. (A) The phase and amplitude of the low frequency signal (blue) modulate the probability of a high frequency spike (orange). (B) The surfaces $S_{A_{\text{low}}}$ (red) and $S_{A_{\text{low}}, \phi_{\text{low}}}$ (yellow). The phase of maximal $A_{\text{high}}$ modulation depends on $A_{\text{low}}$. (C) The modulation index fails to detect this type of PAC.
DOI: https://doi.org/10.7554/eLife.44287.011

non-uniform shape of the $(A_{\text{low}}, \phi_{\text{low}})$ surface is lost when we fail to account for $A_{\text{low}}$. In this scenario, the distribution of $A_{\text{high}}$ over $\phi_{\text{low}}$ appears uniform, resulting in a low **MI** value (*Figure 10C*).

## Application to in vivo human seizure data

To evaluate the performance of the proposed method on in vivo data, we first consider an example recording from human cortex during a seizure (see Materials and methods*: Human subject data*). Visual inspection of the LFP data (*Figure 11A*) reveals the emergence of large amplitude voltage fluctuations during the approximately 80 s seizure. We compute the spectrogram over the entire seizure, using windows of width 0.8 s with 0.002 s overlap, and identify a distinct 10 s interval of increased power in the 4–7 Hz band (*Figure 11B*). We analyze this section of the voltage trace $V$, filtering into $V_{\text{high}}$ (100–140 Hz) and $V_{\text{low}}$ (4–7 Hz), and extracting $A_{\text{high}}$, $A_{\text{low}}$, and $\phi_{\text{low}}$ as in Methods (*Figure 11C*). Visual inspection reveals the occurrence of large amplitude, low frequency oscillations and small amplitude, high frequency oscillations.

We find during this interval significant phase-amplitude coupling computed using $\mathbf{R}_{\text{PAC}}$ ($\mathbf{R}_{\text{PAC}} = 1.55$, $p_{\text{PAC}} = 0.005$, *Figure 12*), and using the modulation index ($\mathbf{MI} = 0.03$, $p_{\text{MI}} = 5.0 \times 10^{-4}$). To examine the phase-amplitude coupling in more detail, we isolate a 2 s segment (*Figure 11D*) and display the signal $V$, the high frequency signal $V_{\text{high}}$, the low frequency phase $\phi_{\text{low}}$, and the low frequency amplitude $A_{\text{low}}$. We observe that when $\phi_{\text{low}}$ is near $\pi$, the amplitude of $V_{\text{high}}$ tends to increase, consistent with the presence of PAC and a significant value of $\mathbf{R}_{\text{PAC}}$ and **MI**.

We also find significant amplitude-amplitude coupling computed using $\mathbf{R}_{\text{AAC}}$ ($\mathbf{R}_{\text{AAC}} = 0.85$, $p_{\text{AAC}} = 0.005$). Comparing $A_{\text{high}}$ and $A_{\text{low}}$ over the 10 s interval (each smoothed using a 1 s moving average filter and normalized), we observe that both $A_{\text{high}}$ and $A_{\text{low}}$ steadily increase over the duration of the interval (*Figure 11E*).

## Application to in vivo rodent data

As a second example to illustrate the performance of the new method, we consider LFP recordings from from the infralimbic cortex (IL) and basolateral amygdala (BLA) of an outbred Long-Evans rat before and after the delivery of an experimental electrical stimulation intervention described in *Blackwood et al. (2018)*. Eight microwires in each region, referenced as bipolar pairs, sampled the LFP at 30 kHz, and electrical stimulation was delivered to change inter-regional coupling (see *Blackwood et al., 2018* for a detailed description of the experiment). Here we examine how cross-frequency coupling between low frequency (5–8 Hz) IL signals and high frequency (70–110 Hz) BLA signals changes from the pre-stimulation to the post-stimulation condition. To do so, we filter the data $V$ into low and high frequency signals (see Materials and methods), and compute the **MI**, $\mathbf{R}_{\text{PAC}}$ and $\mathbf{R}_{\text{AAC}}$ between each possible BLA-IL pairing, sixteen in total.

We find three separate BLA-IL pairings where $\mathbf{R}_{\text{PAC}}$ reports no significant PAC pre- or post-stimulation, but **MI** reports significant coupling post-stimulation. Investigating further, we note that in all three cases, the amplitude of the low frequency IL signal increases from pre- to post-stimulation, and $\mathbf{R}_{\text{AAC}}$, the measure of amplitude-amplitude coupling, increases from pre- to post-stimulation. These observations are consistent with the simulations in *Results: The proposed method is less affected by fluctuations in low-frequency amplitude and AAC*, in which we showed that increases in the low frequency amplitude and AAC produced increases in **MI**, although the PAC remained fixed. We therefore propose that, consistent with these simulation results, the increase in **MI** observed in these data may result from changes in the low frequency amplitude and AAC, not in PAC.

## Using the flexibility of GLMs to improve detection of phase-amplitude coupling in vivo

One advantage of the proposed framework is its flexibility: covariates are easily added to the generalized linear model and tested for significance. For example, we could include covariates for trial, sex, and stimulus parameters and explore their effects on PAC, AAC, or both.

Here, we illustrate this flexibility through continued analysis of the rodent data. We select a single electrode recording from these data, and hypothesize that the condition, either pre-stimulation or post-stimulation, affects the coupling. To incorporate this new covariate into the framework, we consider the concatenated voltage recordings from the pre-stimulation condition $V_{\text{pre}}$ and the post-stimulation condition $V_{\text{post}}$:

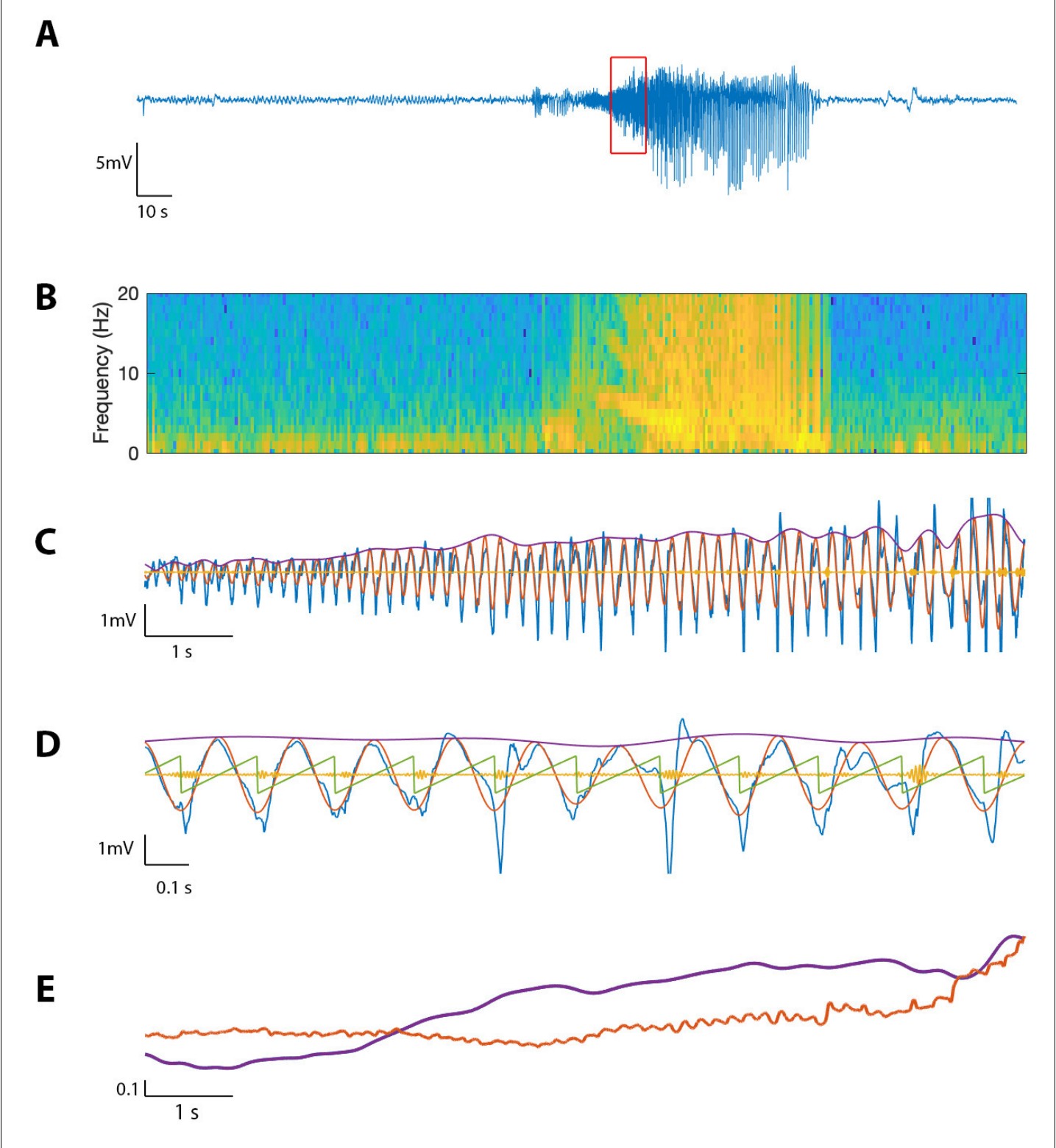

**Figure 11.** The proposed method detects cross-frequency coupling in an in vivo human recording. (**A,B**) Voltage recording (**A**) and spectrogram (**B**) from one MEA electrode over the course of a seizure; PAC and AAC were computed for the time segment outlined in red. (**C**) The 10 s voltage trace (blue) corresponding to the outlined segment in (**A**), and $V_{low}$ (red), $V_{high}$ (yellow), and $A_{low}$ (purple). (**D**) A 2 s subinterval of the voltage trace (blue), $V_{low}$ (red), $V_{high}$ (yellow), $A_{low}$ (purple), and $\phi_{low}$ (green). (**E**) $A_{low}$ (purple) and $A_{high}$ (red) for the 10 s segment in (**C**), normalized and smoothed.
DOI: https://doi.org/10.7554/eLife.44287.012

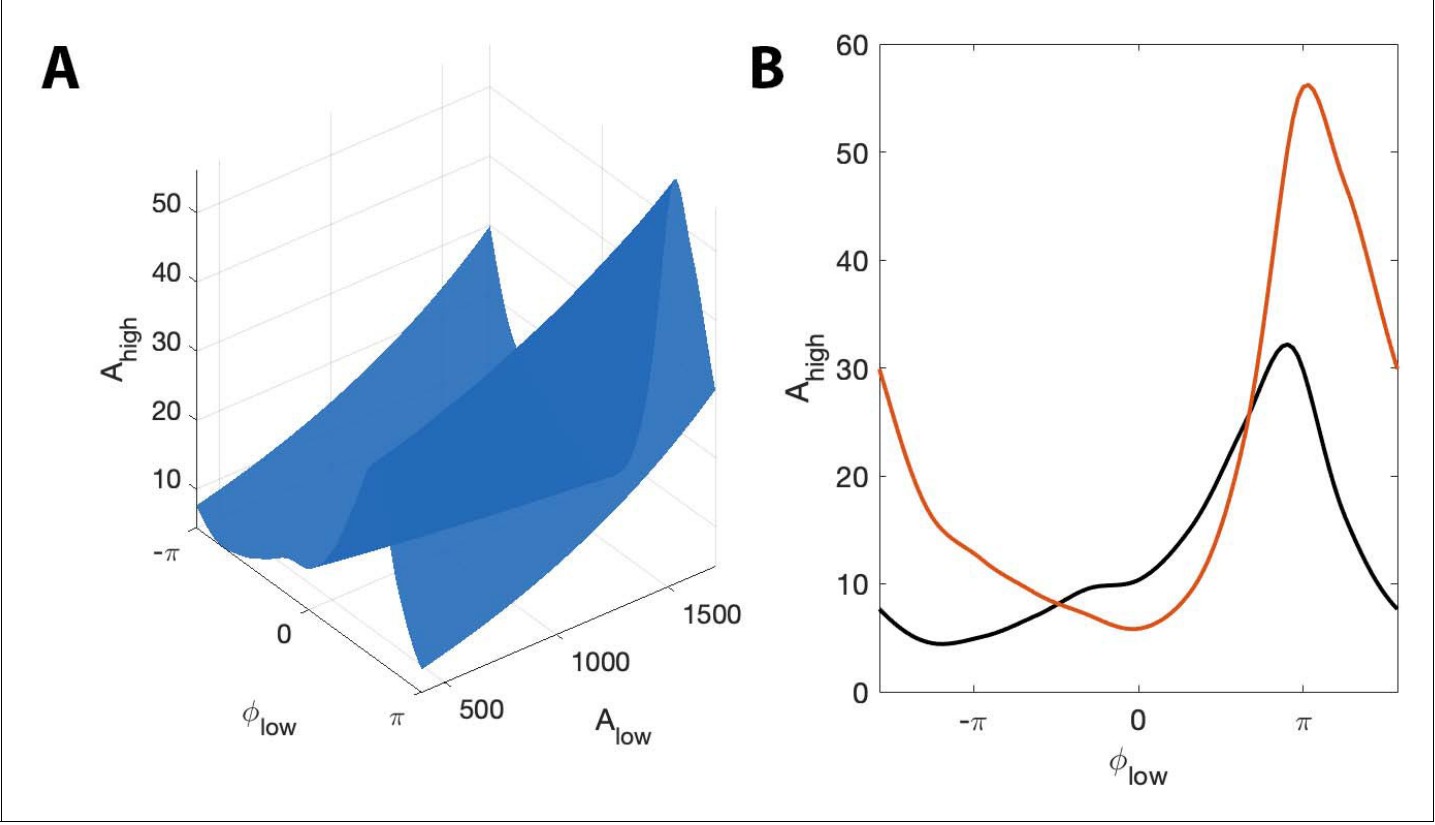

**Figure 12.** The $S_{A_{\mathrm{low}},\phi_{\mathrm{low}}}$ surface shows how PAC changes with the low frequency amplitude and phase during an interval of human seizure. (A) The full model surface (blue) in the ($\phi_{\mathrm{low}}$, $A_{\mathrm{low}}$, $A_{\mathrm{high}}$) space, and components of that surface when (B) $A_{\mathrm{low}}$ is small (black), and $A_{\mathrm{low}}$ is large (red).

DOI: https://doi.org/10.7554/eLife.44287.013

$$V = [V_{\mathrm{pre}}, V_{\mathrm{post}}].$$

From $V$, we obtain the corresponding high frequency signal $V_{\mathrm{high}}$ and low frequency signal $V_{\mathrm{low}}$, and subsequently the high frequency amplitude $A_{\mathrm{high}}$, low frequency phase $\phi_{\mathrm{low}}$, and low frequency amplitude $A_{\mathrm{low}}$. We use these data to generate two new models:

$$A_{\mathrm{high}}|\phi_{\mathrm{low}}, A_{\mathrm{low}}, P \sim \mathrm{Gamma}[\mu, \nu], \tag{9}$$

$$\log \mu = \sum_{k=1}^{n} \beta_k f_k(\phi_{\mathrm{low}}) + \beta_{n+1} A_{\mathrm{low}} + \beta_{n+2} A_{\mathrm{low}} \sin(\phi_{\mathrm{low}}) + \beta_{n+3} A_{\mathrm{low}} \cos(\phi_{\mathrm{low}}) + P(\sum_{j=1}^{n} \beta_{n+3+j} f_j(\phi_{low}) + \beta_{2n+4} A_{low})$$

$$A_{\mathrm{high}}|\phi_{\mathrm{low}}, A_{\mathrm{low}}, P \sim \mathrm{Gamma}[\mu, \nu] \tag{10}$$

$$\log \mu = \sum_{k=1}^{n} \beta_k f_k(\phi_{\mathrm{low}}) + \beta_{n+1} A_{\mathrm{low}} + \beta_{n+2} A_{\mathrm{low}} \sin(\phi_{\mathrm{low}}) + \beta_{n+3} A_{\mathrm{low}} \cos(\phi_{\mathrm{low}}) + P(\beta_{n+4} A_{low}),$$

where $P$ is an indicator function specifying whether the signal is in the *pre-stimulation* ($P = 0$) or *post-stimulation* ($P = 1$) condition. The effect of the indicator function is to include the effect of stimulus condition on the high frequency amplitude. The models in *Equations 9 and 10* now include the effect of low frequency amplitude, low frequency phase, and condition on high frequency amplitude. To determine whether the condition has an effect on PAC, we test whether the term $P(\sum_{j=1}^{n} \beta_{n+3+j} f_j(\phi_{\mathrm{low}}))$ in *Equation 9* is significant, that is whether there is a significant difference

between the models in *Equations 9 and 10*. If the difference between the two models is very small, we gain no improvement in modeling $A_{\text{high}}$ by including the interaction between $P$ and $\phi_{\text{low}}$. In that case, the impact of $\phi_{\text{low}}$ on $A_{\text{high}}$ can be modeled without considering stimulus condition $P$, that is the impact of stimulus condition on PAC is negligible.

To measure the difference between the models in *Equations 9 and 10*, we construct a surface $S_{P\phi_{\text{low}}}$ from the model in *Equation 9*, and a surface $S_P$ from the model in *Equation 10* in the $(A_{\text{low}}, \phi_{\text{low}}, A_{\text{high}}, P)$ space, assessing the models at $P = 1$. We compute $\mathbf{R}_{\text{PAC, condition}}$, which measures the impact of stimulus condition on PAC, as:

$$\mathbf{R}_{\text{PAC, condition}} = \max\left[\text{abs}[1 - S_P/S_{P\phi_{\text{low}}}]\right]. \tag{11}$$

We find for the example rodent data an $\mathbf{R}_{\text{PAC, condition}}$ value of 0.3608, with a p-value of 0.0005. Hence, we find evidence for a significant effect of stimulus on PAC.

To further explore this assessment of stimulus condition on PAC, we simulate 1000 instances of a 40 s signal divided into two conditions: no PAC for the first 20 s ($I_{\text{PAC}} = 0$) and non-zero PAC for the final 20 s ($I_{\text{PAC}} = 1$). We design this simulation to mimic an increase in PAC from *pre-stimulation* to *post-stimulation* (*Figure 13A*). Using the models in *Equations 9 and 10*, and computing $\mathbf{R}_{\text{PAC, condition}}$, we find $p<0.05$ for 100% of simulated signals. We also simulate 1000 instances of a 40 s signal with no PAC ($I_{\text{PAC}} = 0$) for the entire 40 s, that is PAC does not change from *pre-stimulation* to *post-stimulation* (*Figure 13B*), and find in this case $p<0.05$ for only 4.6% of simulations. Finally, we simulate 1000 instances of a 40 s signal with fixed PAC ($I_{\text{PAC}} = 1$), and with a doubling of the low frequency amplitude occuring at 20 s (i.e., pre-stimulation the low frequency amplitude is 1, and

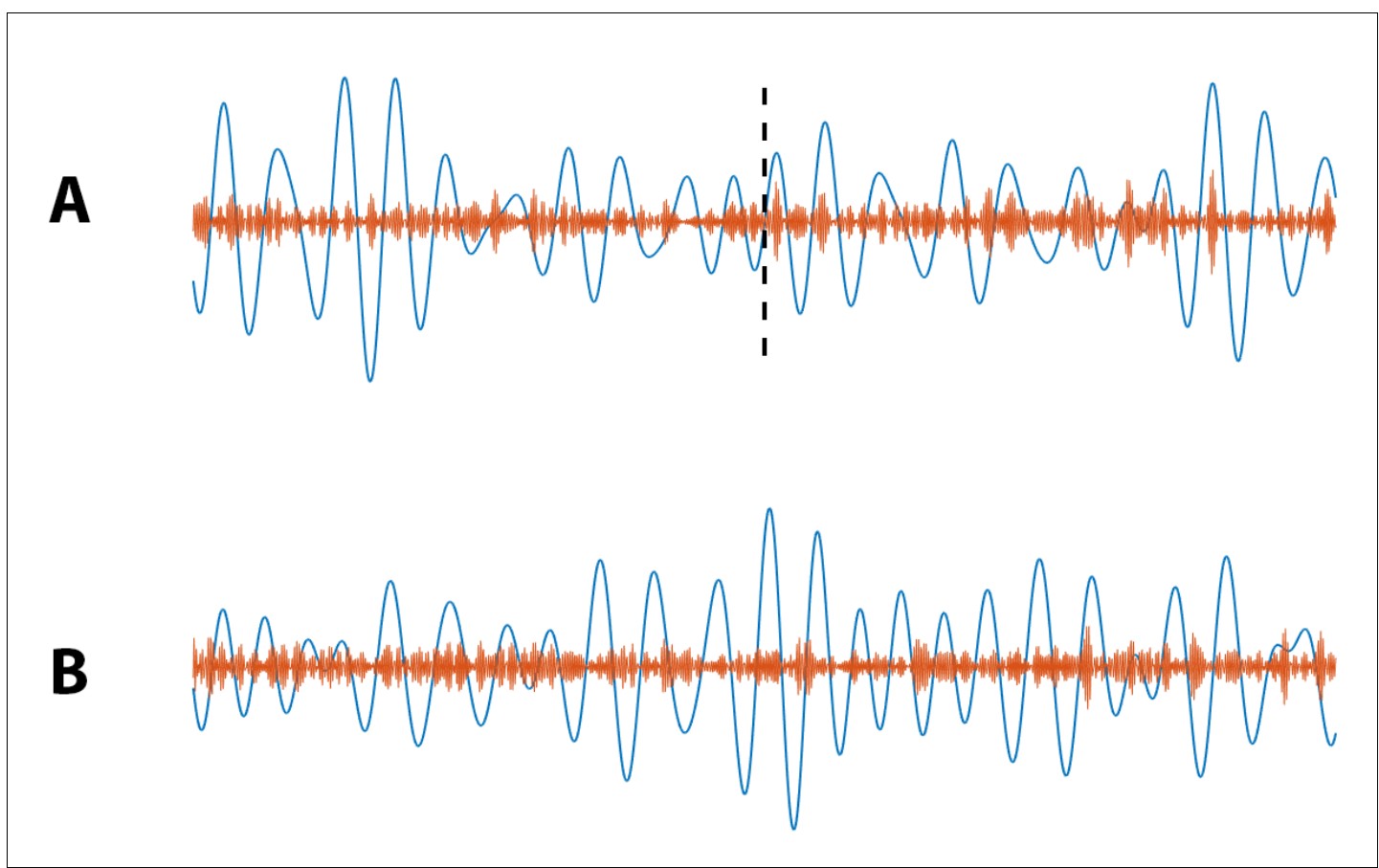

**Figure 13.** Example simulated $V_{\text{low}}$ (blue) and $V_{\text{high}}$ (orange) signals for which (**A**) PAC increases at 20 s (indicated by black dashed line), and (**B**) no increase in PAC occurs.
DOI: https://doi.org/10.7554/eLife.44287.014

post-stimulation the low frequency amplitude is 2). We find $p<0.05$ for only 3.6% of simulations. We conclude that this method effectively determines whether stimulation condition significantly changes PAC.

This example illustrates the flexibility of the statistical modeling framework. Extending this framework is straightforward, and new extensions allow a common principled approach to test the impact of new predictors. Here we considered an indicator function that divides the data into two states (pre- and post-stimulation). We note that the models are easily extended to account for multiple discrete predictors such as gender and participation in a drug trial, or for continuous predictors such as age and time since stimulus.

## Discussion

In this paper, we proposed a new method for measuring cross-frequency coupling that accounts for both phase-amplitude coupling and amplitude-amplitude coupling, along with a principled statistical modeling framework to assess the significance of this coupling. We have shown that this method effectively detects CFC, both as PAC and AAC, and is more sensitive to weak PAC obscured by or coupled to low-frequency amplitude fluctuations. Compared to an existing method, the modulation index (*Tort et al., 2010*), the newly proposed method more accurately detects scenarios in which PAC is coupled to the low-frequency amplitude. Finally, we applied this method to in vivo data to illustrate examples of PAC and AAC in real systems, and show how to extend the modeling framework to include a new covariate.

One of the most important features of the new method is an increased ability to detect weak PAC coupled to AAC. For example, when sparse PAC events occur only when the low frequency amplitude ($A_{\mathrm{low}}$) is large, the proposed method detects this coupling while another method not accounting for $A_{\mathrm{low}}$ misses it. While PAC often occurs in neural data, and has been associated with numerous neurological functions (*Canolty and Knight, 2010*; *Hyafil et al., 2015b*), the simultaneous occurrence of PAC and AAC is less well studied (*Osipova et al., 2008*). Here, we showed examples of simultaneous PAC and AAC recorded from human cortex during a seizure, and we note that this phenomena has been simulated in other works (*Mazzoni et al., 2010*).

While the exact mechanisms that support CFC are not well understood (*Hyafil et al., 2015b*), the general mechanisms of low and high frequency rhythms have been proposed. Low frequency rhythms are associated with the aggregate activity of large neural populations and modulations of neuronal excitability (*Engel et al., 2001*; *Varela et al., 2001*; *Buzsáki and Draguhn, 2004*), while high frequency rhythms provided a surrogate measure of neuronal spiking (*Rasch et al., 2008*; *Mukamel et al., 2005*; *Fries et al., 2001*; *Pesaran et al., 2002*; *Whittingstall and Logothetis, 2009*; *Ray and Maunsell, 2011*; *Ray et al., 2008b*). These two observations provide a physical interpretation for PAC: when a low frequency rhythm modulates the excitability of a neural population, we expect spiking to occur (i.e., an increase in $A_{\mathrm{high}}$) at a particular phase of the low frequency rhythm ($\phi_{\mathrm{low}}$) when excitation is maximal. These notions also provide a physical interpretation for AAC: increases in $A_{\mathrm{low}}$ produce larger modulations in neural excitability, and therefore increased intervals of neuronal spiking (i.e., increases in $A_{\mathrm{high}}$). Alternatively, decreases in $A_{\mathrm{low}}$ reduce excitability and neuronal spiking (i.e., decreases in $A_{\mathrm{high}}$).

The function of concurrent PAC and AAC, both for healthy brain function and during a seizure as illustrated here, is not well understood. As PAC occurs normally in healthy brain signals, for example during working memory, neuronal computation, communication, learning and emotion (*Tort et al., 2009*; *Jensen et al., 2016*; *Canolty and Knight, 2010*; *Dejean et al., 2016*; *Karalis et al., 2016*; *Likhtik et al., 2014*; *Jones and Wilson, 2005*; *Lisman, 2005*; *Sirota et al., 2008*), these preliminary results may suggest a pathological aspect of strong AAC occurring concurrently with PAC.

Proposed functions of PAC include multi-item encoding, long-distance communication, and sensory parsing (*Hyafil et al., 2015b*). Each of these functions takes advantage of the low frequency phase, encoding different objects or pieces of information in distinct phase intervals of $\phi_{\mathrm{low}}$. PAC can be interpreted as a type of focused attention; $A_{\mathrm{high}}$ modulation occurring only in a particular interval of $\phi_{\mathrm{low}}$ organizes neural activity - and presumably information - into discrete packets of time. Similarly, a proposed function of AAC is to encode the number of represented items, or the amount of information encoded in the modulated signal (*Hyafil et al., 2015b*). A pathological increase in AAC may support the transmission of more information than is needed, overloading the

communication of relevant information with irrelevant noise. The attention-based function of PAC, that is having reduced high frequency amplitude at phases not containing the targeted information, may be lost if the amplitude of the high frequency oscillation is increased across wide intervals of low frequency phase.

Like all measures of CFC, the proposed method possesses specific limitations. We discuss five limitations here. First, the choice of spline basis to represent the low frequency phase may be inaccurate, for example if the PAC changes rapidly with $\phi_{\text{low}}$. Second, the value of $\mathbf{R}_{\text{AAC}}$ depends on the range of $A_{\text{low}}$ observed. This is due to the linear relationship between $A_{\text{low}}$ and $A_{\text{high}}$ in the $A_{\text{low}}$ model, which causes the maximum distance between the surfaces $S_{A_{\text{low}}}$ and $S_{A_{\text{low}},\phi_{\text{low}}}$ to occur at the largest or smallest value of $A_{\text{low}}$. To mitigate the impact of extreme $A_{\text{low}}$ values on $\mathbf{R}_{\text{AAC}}$, we evaluate the surfaces $S_{A_{\text{low}}}$ and $S_{A_{\text{low}},\phi_{\text{low}}}$ over the 5th to 95th quantiles of $A_{\text{low}}$. We note that an alternative metric of AAC could instead evaluate the slope of the $S_{A_{\text{low}}}$ surface; to maintain consistency of the PAC and AAC measures, we chose not to implement this alternative measure here. Third, the frequency bands for $V_{\text{high}}$ and $V_{\text{low}}$ must be established before $\mathbf{R}$ values are calculated. Hence, if the wrong frequency bands are chosen, coupling may be missed. It is possible, though computationally expensive, to scan over all reasonable frequency bands for both $V_{\text{high}}$ and $V_{\text{low}}$, calculating $\mathbf{R}$ values for each frequency band pair. Fourth, we note that the proposed modeling framework assumes the data contain approximately sinusoidal signals, which have been appropriately isolated for analysis. In general, CFC measures are sensitive to non-sinusoidal signals, which may confound interpretation of cross-frequency analyses (*Cole and Voytek, 2017*; *Kramer et al., 2008*; *Aru et al., 2015*). While the modeling framework proposed here does not directly account for the confounds introduced by non-sinusoidal signals, the inclusion of additional predictors (e.g. detections of sharp changes in the unfiltered data) in the model may help mitigate these effects. Fifth, we simulate time series with known PAC and AAC, and then test whether the proposed analysis framework detects this coupling. The simulated relationships between $A_{\text{high}}$ and $(\phi_{\text{low}}, A_{\text{low}})$ may result in time series with simpler structure than those observed in vivo. For example, a latent signal may drive both $A_{\text{high}}$ and $\phi_{\text{low}}$, and in this way establish nonlinear relationships between the two observables $A_{\text{high}}$ and $\phi_{\text{low}}$. We note that, if this were the case, the latent signal could also be incorporated in the statistical modeling framework (*Yousefi et al., 2019*).

We chose the statistics $\mathbf{R}_{\text{PAC}}$ and $\mathbf{R}_{\text{AAC}}$ for two reasons. First, we found that two common methods of model comparison for GLMs provide less robust measures of significance than $\mathbf{R}_{\text{PAC}}$ and $\mathbf{R}_{\text{AAC}}$. While the statistics $\mathbf{R}_{\text{PAC}}$ and $\mathbf{R}_{\text{AAC}}$ are less powerful than standard model comparison tests, the large amount of data typically assessed in CFC analysis may compensate for this loss. We showed that the statistics $\mathbf{R}_{\text{PAC}}$ and $\mathbf{R}_{\text{AAC}}$ performed well in simulations, and we note that these statistics are directly interpretable. While many model comparison methods exist - and another method may provide specific advantages - we found that the framework implemented here is sufficiently powerful, interpretable, and robust for real-world neural data analysis.

The proposed method can easily be extended by inclusion of additional predictors in the GLM. Polynomial $A_{\text{low}}$ predictors, rather than the current linear $A_{\text{low}}$ predictors, may better capture the relationship between $A_{\text{low}}$ and $A_{\text{high}}$. One could also include different types of covariates, for example classes of drugs administered to a patient, or time since an administered stimulus during an experiment. To capture more complex relationships between the predictors ($A_{\text{low}}$, $\phi_{\text{low}}$) and $A_{\text{high}}$, the GLM could be replaced by a more general form of Generalized Additive Model (GAM). Choosing GAMs would remove the restriction that the conditional mean $A_{\text{high}}$ must be linear in each of the model parameters (which would allow us to estimate knot locations directly from the data, for example), at the cost of greater computational time to estimate these parameters. The code developed to implement the method is flexible and modular, which facilitates modifications and extensions motivated by the particular data analysis scenario. This modular code, available at https://github.com/Eden-Kramer-Lab/GLM-CFC, also allows the user to change latent assumptions, such as choice of frequency bands and filtering method. The code is freely available for reuse and further development.

Rhythms, and particularly the interactions of different frequency rhythms, are an important component for a complete understanding of neural activity. While the mechanisms and functions of some rhythms are well understood, how and why rhythms interact remains uncertain. A first step in addressing these uncertainties is the application of appropriate data analysis tools. Here we provide a new tool to measure coupling between different brain rhythms: the method utilizes a statistical

modeling framework that is flexible and captures subtle differences in cross-frequency coupling. We hope that this method will better enable practicing neuroscientists to measure and relate brain rhythms, and ultimately better understand brain function and interactions.

## Acknowledgements

This work was supported in part by the National Science Foundation Award #1451384, in part by R01 EB026938, in part by R21 MH109722, and in part by the National Science Foundation (NSF) under a Graduate Research Fellowship.

## Additional information

### Funding

| Funder | Grant reference number | Author |
| --- | --- | --- |
| National Science Foundation | NSF DMS #1451384 | Jessica K Nadalin<br>Mark A Kramer |
| National Science Foundation | GRFP | Jessica K Nadalin |
| National Institutes of Health | R21 MH109722 | Alik S Widge |
| National Institutes of Health | R01 EB026938 | Alik S Widge<br>Uri T Eden<br>Mark A Kramer |

The funders had no role in study design, data collection and interpretation, or the decision to submit the work for publication.

### Author contributions

Jessica K Nadalin, Conceptualization, Software, Formal analysis, Validation, Investigation, Visualization, Methodology, Writing—original draft, Writing—review and editing; Louis-Emmanuel Martinet, Data curation, Writing—review and editing; Ethan B Blackwood, Meng-Chen Lo, Resources, Data curation, Software; Alik S Widge, Resources, Investigation, Writing—review and editing; Sydney S Cash, Investigation, Writing—review and editing; Uri T Eden, Software, Methodology, Writing—review and editing; Mark A Kramer, Conceptualization, Software, Supervision, Methodology, Writing—review and editing

### Author ORCIDs

Ethan B Blackwood http://orcid.org/0000-0002-3049-0640
Meng-Chen Lo http://orcid.org/0000-0003-3913-3233
Alik S Widge http://orcid.org/0000-0001-8510-341X
Mark A Kramer https://orcid.org/0000-0002-9979-7202

### Ethics

Human subjects: All patients were enrolled after informed consent, and consent to publish, was obtained and approval was granted by local Institutional Review Boards at Massachusetts General Hospital and Brigham Women's Hospitals (Partners Human Research Committee), and at Boston University according to National Institutes of Health guidelines (IRB Protocol # 1558X).
Animal experimentation: The animal experimentation received IACUC approval from the University of Minnesota (IACUC Protocol # 1806-36024A).

### Decision letter and Author response

Decision letter https://doi.org/10.7554/eLife.44287.017
Author response https://doi.org/10.7554/eLife.44287.018

## Additional files

### Supplementary files

• Transparent reporting form DOI: https://doi.org/10.7554/eLife.44287.015

### Data availability

In vivo human data available at https://github.com/Eden-Kramer-Lab/GLM-CFC (copy archived at https://github.com/elifesciences-publications/GLM-CFC). In vivo rat data available at https://github.com/tne-lab/cl-example-data (copy archived at https://github.com/elifesciences-publications/cl-example-data).

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
