## [Decision Letter]

[Editors’ note: the authors were asked to provide a plan for revisions before the editors issued a final decision. What follows is the editors’ letter requesting such plan.]

Thank you for sending your article entitled "A statistical modeling framework to assess cross-frequency coupling while accounting for confounding effects" for peer review at *eLife*. Your article is being evaluated by Laura Colgin as the Senior Editor, a Reviewing Editor, and three reviewers.

Given the list of essential revisions, the editors and reviewers invite you to respond with an action plan and timetable for the completion of the additional work. We plan to share your responses with the reviewers and then issue a binding recommendation.

While the work was appreciated (that is, the importance of including statistical approaches), several aspects were raised that require clarification and additional work (e.g., comparison to other methods, confounding factor issues, etc.). It is unclear whether this would be addressable by the authors and in a timely fashion. As this is understood to be a methods paper, it was not deemed critical for the authors to provide explanations per se of the meaning of results, although they would be welcome to do so.

Essential revisions:

The article by Jessica Nadalin and colleagues makes a key improvement in the statistical methods to detect cross-frequency coupling in neural signals. As the coupling between neural oscillations has become the focus of intense research and has been linked to a wide array of cognitive functions, the proposed method has very general implications for neuroscience. It may constitute the first attempt to assess conjointly different types of CFC. The use of generative models, in particular GLMs, seems like a solid way for building complex statistical approaches – they are increasingly popular in neuroscience to analyze behavioral and neural spiking data. There are nevertheless quite a number of points that should be resolved or clarified in the manuscript to provide a clear validation of the method.

1) The model for PAC and CFC is a Generalized Additive Model (GAM) rather than a GLM, as it takes the form: log *μ* = f(*Ф*_low_).

Here the function f is approximated by splines, which is one typical decomposition of GAMs. See the relevant literature, e.g. textbook by Wood, which provides principled ways of selecting the number of splines, assessing the uncertainty about the inferred function f, performing model comparison, etc. In particular it would be nice to show a few examples of the estimated function f, the typical modulation of *A*_high_ by *Ф*_low_.

2) Presumably, the formula for R is an attempt to mimic the R^2^ metrics used in linear regression. However there already exists of series of pseudo-R^2^ metrics for GLMs, for which pros and cons have been discussed, see e.g. https://web.archive.org/web/20130701052120/http://www.ats.ucla.edu:80/stat/mult_pkg/faq/general/Psuedo_RSquareds.htm. These measures should not suffer from the problem of using an unbounded space as is the case here for R_AAC_ and R_CFC._ The authors should pick one of these and apply it instead of the R measure (unless authors provide a clear explanation of why they used such definition of R).

3) Comparing statistically CFC values between distinct conditions is a key experimental method. The method described in rat data looks really nice, but it would need to be assessed beforehand on synthetic data. Will the method display false alarms (incorrect detection of changes in CFC) when the overall amplitude of the low frequency changes between conditions, as would direct comparison of CFC values between conditions? A method that gets rid of this confound would be a major advancement. Moreover, authors fail to explain the discrepancy in rat data between MI results and GLM results. Is it because of variations of the amplitude of the low frequency signal between conditions?

4) The authors quite surprisingly test their different methods of computing p-values of different synthetic datasets, and never on the same. It is frustrating that one cannot conclude in the end which method provides better results. Moreover, the two methods for generic synthetic data should be presented next to each other to allow comparison (the presentations of the equations could be made more similar).

5) It is important to see how the GLM method compares to other PAC detection method (e.g. MI) for weaker modulations. Is it as sensitive? Moreover, reviewers felt that the comparison of the sensitivity of the PAC measures to changes in *A*_low_ (Figure 5) was quite unfair. MI is an unbounded measure while R cannot be larger than 1. If reviewers understood correctly, the base value for scale factor should be around 0.5 (from Figure 4F), so that it cannot increase more than a twofold increase. This could explain the plateau in Figure 5. Is this correct? Perhaps performing the same analysis with a much weaker coupling would remove this concern.

6) The modulation of PAC by the low-frequency amplitude is an important contribution of the paper. It would deserve an actual figure for the second patient data, illustrating it with the method and showing modulations of *A*_high_ by *Ф*_low_ splitting the dataset into low and high *A*_low_. From a mechanistic point of view, the fact that PAC is larger when *A*_low_ is larger makes perfect sense, and it has been linked to the generation of AAC (Hyafil et al., 2015 Figure 3). Is there an intuitive generative mechanism that would create lower PAC for larger *A*_low_ (as in Figure 7)?

7) Figure 9 and Figure 10 make apparent that the recovered amplitude of the low frequency signal *A*_low_ fluctuates within each cycle, which goes against the very notion of amplitude of an oscillation. This can lead to falsely detecting AAC when there is PAC, as is evident for example in Figure 9: *A*_high_ is higher at the phase of the low-frequency signal where *A*_low_ is higher. Proper AAC should mean that *A*_high_ is higher for entire cycles of low frequency where *A*_low_ is high/low.

8) Clarification of confounding aspect intention and circularity perception in the method:

8.1) The simulation part is suboptimal and to some extent based on circularity. Specifically, the method is based on modelling the Hilbert-envelope of high frequency band limited activity with different GLMs, using regressors that are functions of the Hilbert envelope and phase of low frequency band limited activity. The test statistic is derived from the response functions, and statistical inference can be done with parametric methods, or, more appropriately for real data, using bootstrap techniques.

For the simulations, a generative model was used which essentially builds the high frequency amplitude component as a function of low frequency phase component and/or the low frequency amplitude component. Thus, it is not really surprising that the method works, specifically if in the processing and generation of the data similar filter passbands etc. have been used. It is unclear how this can be alleviated, unless the perceived circularity is actually not there, or the authors are able to come up with a fairer way to simulate the data in order to convincingly show that their method works in general.

8.2) Next, in support of the claim that the proposed test-statistic 'accounts for confounding effects', I feel that the evidence presented at most shows that the R_CFC_ metric scales less strongly with low frequency amplitude (Figure 5), this property should not be oversold.

8.3) With respect to the application to real data, the abstract mentions that "we illustrate how CFC evolves during seizures and is affected by electrical stimuli". This indeed is illustrated by the data, in that the CFC metric is modulated. Yet, it is questionable (specifically for the human ictal data) that this reflects (patho)physiologically meaningful interactions between band-limited neural signal components. If anything, the increased CFC metric highlights the highly non-sinusoidal nature of the ictal spikes, and demonstrates the generic sensitivity of CFC-measures to the non-sinusoidality of the associated periodic signal components. This is a well-known feature, and the most important confounding interpretational issue in most cross-frequency analyses. Therefore, the high expectations based on the manuscript's title were not met in that respect. This important issue needs to be discussed in more detail, and the title should be adjusted for the sake of the reader's expectation management.

9) The authors proposed new measures of cross frequency coupling (CFC) in neurophysiological data where the emphasis is placed on phase-amplitude coupling (PAC) and amplitude-amplitude coupling (AAC). Statistical properties of the new estimators are discussed. These methods are tested in simulated data and in neural recordings from human epilepsy patients and rodents undergoing electrical stimulation. Whereas CFC is an important research area, and better measures are always welcome, it was not clear if the proposed methods are sufficiently novel to potentially offer new physiological insights into the neural operations represented by the data.

Estimating phase and amplitude from Hilbert transform requires the signal to be narrow band. Whereas the 4-7 Hz filtering band for theta phase and amplitude estimation is adequate, the 100-140 Hz filtering band for estimating high gamma phase and amplitude is too wide; it is thus likely that the phase and amplitude estimated this way is not accurate. A way to empirically test whether the filtering band is sufficiently narrow is to plot the real and the imaginary part of the filtered signal in a phase portrait to see whether the trajectory moves around a well-formed center. It is the rotation around this center that allows us to define instantaneous phase and amplitude properly from Hilbert transforms.

Numerous ad hoc assumptions go into Equations (1)-(4). It is difficult to follow the logic behind the method. The beauty of the original CFC measures (e.g., Canolty's or Tort's definition of PAC) is that they are simple and intuitive.

The synthetic time series considered here are not biologically realistic. First, spiking neural models should be used to generate such time series. In particular, the model should incorporate the property that gamma and high gamma in some way reflect neural spiking. Second, noise should be added to the synthetic time series to mimic the real world recordings; the amplitude of the noise can be used to assess the influence of signal to noise ratio on the various CFC quantities defined.

For the human seizure data, the authors only evaluated their new measures on the data, but did not compute the commonly applied PAC measures for comparison.

[Editors' note: further revisions were requested prior to acceptance, as described below.]

Thank you for resubmitting your work entitled "A statistical framework to assess cross-frequency coupling while accounting for modeled confounding effects" for further consideration at *eLife*. Your revised article has been favorably evaluated by Laura Colgin (Senior Editor), a Reviewing Editor (Frances Skinner), and 3 reviewers (Reviewer #1: Alexandre Hyafil; Reviewer #2: Jan-Mathijs Schoffelen; Reviewer #3 has opted to remain anonymous).

The reviewers appreciated the new figures and improvements throughout. The manuscript is likely to be accepted after the remaining essential issues below are addressed; however, they still need to be addressed. The main thrust of most of these requests is to ensure clarity for readers of where this approach is situated relative to others, and in some cases, there may have been misinterpretations of previous requests.

1) Title: Another suggestion is that one could use "analysis", rather than "modeling" to have a title of "… confounding analysis effects". This is a suggestion, not required.

2) The new Figure 2 is fine. It was thought that the comment regarding GLM vs. GAM was misinterpreted. The point was conceptual rather than algorithmic. The essence of the models described here is to capture a nonlinear mapping of the regressor(s), that is then transformed into the observable using a link function and some observation noise: this is exactly what a GAM is. As described in the textbook by Wood (section 4.2), using basis functions such as splines, a GAM can be transformed into a (penalized) GLM, and then exactly fit using the standard GLM procedure that authors have used. (I believe outfitting is rather an outdated form of fitting GAMs). Please simply add some wording to make this clear. That is, the suggestion is not to change the core model fitting part, only to acknowledge the direct link with GAM.

3) We understand that the R measure captures how strongly the regressors modulate high frequency amplitude: this is exactly what R^2^ does in linear regression (assessing the part of the variance explained by the model), although it is sometimes described incorrectly as measuring goodness-of-fit. In their response, reviewers claim a somehow different thing, that their measure "estimate differences between fits from two different models". Now, if this refers to model comparison, again, there is large literature on how to perform model comparison between GLMs (including AIC, cross-validation, etc.). So I can see no reason to create a new method that suffers some important drawbacks (not derived from any apparent principle, arbitrariness for unbounded regressors, PAC measure is modulated by levels of AAC and low frequency amplitude, etc.) and has not been tested against those standard measures. We fully support the idea of using GLM as statistical tools for complex signals; it is a pity though not to leverage on the rich tools that have been developed within the GLM framework (and beyond that in machine learning). Please either use these rich tools or provide a proper argumentation of why your measure is used.

4) The new simulations represent an important step in the good direction but the analysis performed on the synthetic data is not quite the one performed on the experimental data. Figure 7 shows that the R measure is modulated by amplitude of the low frequency signal as well as AAC (although less so than the MI measure), preventing any direct comparison of PAC between differing conditions. Now on the rodent data, authors very astutely use a single model for both conditions, and indicator functions (Equation 10). For some reason, the model comparison performed on the previous version of the manuscript has disappeared. This was viewed as regrettable as this indicated that PAC was modulated between the two experimental conditions, something that direct comparison of MI/R values cannot afford to test. This is also the manipulation that we would like to see tested on synthetic data, to make sure whether authors are using a method to isolate modulations of PAC from modulations of low frequency amplitude and AAC.

In other words, it would be great to see this method tested on synthetic data to see if they get rid of the confounds, and then possibly applied on rat data as in previous manuscript. This was viewed as something that could make the paper much better. However, it is acceptable to also choose to trim the comparison between conditions part. If so, the authors are requested to be very explicit in their wording that by using separate models for different conditions, they cannot get rid of possible confounds.

5) The R measure seems pretty much as sensitive as the MI measure to low PAC. However, the two measures cannot be compared directly. Since most research is geared towards assessing significance levels of PAC, it would be interesting to rather show how the two methods compare in terms of statistical power: comparing type I error rates for low levels of PAC (although reviewers are a bit wary of what could be the results, since the type II error rate related to Figure 5 was less than 1% when it is supposedly calibrated to be 5%, suggesting the method could be too conservative).

6) The response to point 7 misses the important message: the extraction of the amplitude of the low frequency signal is intrinsically flawed as the recovered amplitude fluctuates *within* cycles of the low frequency rhythm. Looking at Figure 9C, peaks of *A*_low_ and *A*_high_ coincide within these cycles, which will very likely give rise to spurious AAC for this segment (irrespective of whether slow fluctuations of *A*_low_ do modulate *A*_high_). It is unclear why this is apparent here but was not detected in the analysis of synthetic data, but in any case, it would require in our opinion a serious assessment of the properties of the algorithm to extract *A*_low_. The authors' solution (changing the frequency range of the simulation to make the problem less apparent) is not a valid one. In any case, we can still see the issue in Figure 11F: sub-second fluctuations in *A*_low_. Perhaps one quick fix would simply be to use a lowpass filter for *A*_low_ to remove these spurious fluctuations. It makes no sense conceptually that a signal assessing the amplitude of an oscillation would fluctuate rapidly within each cycle of that oscillation.

7) Regarding point 8.1 in the rebuttal: the change proposed in generating the signals was viewed as just a minor cosmetic change. It does not take away our concern for circularity, since the high frequency amplitude is still a direct function of the phase time series of the low frequency signal. Also, a change in signal generation was only done for the PAC, and not for the AAC, which suffers from the same circularity concern. As mentioned before, perhaps this concern cannot be alleviated at all (although we think that the authors could have done a better job by generating both the low frequency phase time series *and* the high frequency amplitude time series as a (possibly non-linear) function of a third unobserved time series. Either way, the authors need to at least discuss this concern for circularity explicitly, and argue why this in their view is not a problem in convincingly showing the utility of their new method.

8) Regarding point 8.3 in the rebuttal: we find the proposed fourth concern far too theoretical to be of practical use. The readership really should be informed about the interpretational confounds of CFC metrics, as mentioned in an original comment. We are not convinced that 'appropriate filtering of the data into high and low frequency components' is fundamentally possible, but we'd like to stand corrected with a convincing argumentation. In other words, the authors are requested to be very explicit in the interpretational limitations of any CFC measure, which is independent of the signal processing that is applied to the data before the measure is computed. This is important to be very clear about since 'non-methods' people may use the method without too much thinking about the potential shortcomings.

9) The authors are asked to edit a sentence in their Introduction about cellular mechanisms being ‘well-understood’ for gamma and theta – this was viewed as somewhat arguable for theta. Tort et al. is cited for theta which does not seem to be an appropriate reference for cellular perspectives. A recent paper for cellular mechanisms for theta could be used (Ferguson et al., 2018).

10) Please also consider the following: a) Some of the figures/panels could be placed as figure supplements to avoid distracting the reader away from the main points of the manuscript.

b) The figures could be polished a bit more, notably:

- Use labels such as 'π2' for phase variables.

- Merge single lines panels when possible (e.g. Figure 12B,C).

- Improve the readability of the 3D figures (e.g. using meshgrids; perhaps plotting only the full model surface in Figure 5).

- Adjust font size.

- Adjust axis limit and panel size for better readability (for some panels it's hard to see anything beyond noise, e.g. Figure 4, Figure 9B,C, Figure 10B).

- Figure 5G,K and O are difficult to read.

- Figure 11B: are these stacked bars of is R_PAC_ always larger than R_AAC_? In the former case, it makes R_PAC_ difficult to read: plot unstacked bars or curves instead.

- Figure 11F: use different scales from *A*_high_ and *A*_low_ to make *A*_high_ visible (or normalize signals).

c) Names of the models: why name them “*Ф*_low_” and “*A*_low_” rather than PAC and AAC models?

d) Isn't the constant offset β_0_ missing from Equation 1 and Equation 3?

e) Please use semi-colons instead of commas to separate possible values of x (it took me a while to understand this).

f) In the spiking model, specify that the slow oscillation is imposed externally, not generated by the network.

g) "at the segment indicated by the asterisk in Figure 11B (…)": there is no asterisk in Figure 11B.

h) Figure 11F: " *A*_high_ increases with *A*_low_ over time"-> not clear. *A*_high_ increases over time but it seems that it is higher for intermediate than for high value of *A*_low_.

i) A concern was raised about whether the frequency band corresponding to the low signal in the human data is described in the text.

---

## [Author Response]

Essential revisions:1) The model for PAC and CFC is a Generalized Additive Model (GAM) rather than a GLM, as it takes the form: log μ = f(Ф_low_).

*Here the function f is approximated by splines, which is one typical decomposition of GAMs. See the relevant literature, e.g. textbook by Wood, which provides principled ways of selecting the number of splines, assessing the uncertainty about the inferred function f, performing model comparison, etc. In particular it would be nice to show a few examples of the estimated function f, the typical modulation of A_high_ by Ф*_l*ow*_.

To address these comments, we have updated the manuscript in the following ways. First, we now describe a principled procedure to select the number of splines. We use an AIC-based selection procedure, as described in (Kramer and Eden, 2013). We have updated the manuscript as follows:

“Here, we fix 𝑛 = 10, which is a reasonable choice for smooth PAC with one or two broad peaks (Karalis et at., 2016). To support this choice, we apply an AIC-based selection procedure to 1000 simulated instances of signals of duration 20 s with phase-amplitude coupling and amplitude-amplitude coupling (see Methods: Synthetic Time Series with PAC and Synthetic Time Series with AAC, below, for simulation details). For each simulation, we fit Model 1 to these data for 27 different values of 𝑛 from 𝑛 = 4 to 𝑛 = 30. For each simulated signal, we record the value of 𝑛 such that we minimize the AIC, defined as

AIC = Δ+2𝑛,

where Δ is the deviance from Model 1. The values of 𝑛 that minimize the AIC tend to lie between 𝑛 = 7 and 𝑛 = 12 (Figure 2). These simulations support the choice of 𝑛 = 10 as a sufficient number of splines.”

Second, we note that one purpose of our method is to examine the impact of *Ф*_low_and *A*_low_ on *A*_high_ in 3-dimensional space. However, in response to this comment, we have also updated the manuscript to show examples of the requested projection: the estimated modulation of *A*_high_ by *Ф*_low_. We now include examples of this estimation in Figure 12. Please see our response to comment (6) for examples.

Third, we would like to thank the reviewer for bringing up the distinction between GAMs and GLMs in our approach. We note that although the models [1] and [3] incorporate spline basis functions of low frequency phase as predictors, these models are still GLMs, as the link function of the conditional mean of the response variable (*A*_high)_ varies linearly with all of the model parameters to be estimated. More specifically, we note that the coefficients 𝛽_k_ multiply the splinebasis functions, remaining outside of the functions themselves, consistent with the definition of GLMs. This allows all of the parameters to be estimated directly via an iteratively reweighted least squares procedure, as is common for GLM fitting, as opposed to a more computationally intensive backfitting procedure often used for GAM fitting. We now make this distinction clear in the revised manuscript as follows:

“We use a tension parameter of 0.5, which controls the smoothness of the splines. We note that, because the link function of the conditional mean of the response variable (*A*_high_) varies linearly with the model coefficients 𝛽_k_, the model is a GLM. Here, we fix n=10, which is a reasonable choice […]”

We chose to use GLMs in part for computational efficiency: as noted above, we can fit the GLMs in models [1]-[3] directly using iteratively reweighted least squares, whereas GAMs would require a more complex backfitting algorithm, adding considerable computation time to our method. This computational efficiency is especially important in our approach. For example, to create each surface *S*, we fit a separate GLM 640 times (once for each value of *A*_low_), and to compute *p*-values, we repeat this entire procedure 1000 times. Hence, any small increase in computation time would have a multiplicatively large impact, resulting in a computationally prohibitive measure.

Finally, we note that the primary focus of our method is to determine the impact of predictors (*ɸ*_low_, *A*_low)_ on the response variable *A*_high_ by fitting a full model with functions of both predictors__ and a smaller, nested model with functions of only a single predictor, and comparing the difference in fits between these models. We use this difference to measure the impact of *ɸ*_low_ or__
*A*_low_ on *A*_high_. We have shown in simulation and in our data that the GLMs in models [1]-[3] are sufficiently sensitive to differences in these model fits, detecting even weak impacts of *ɸ*_low_ and__
*A*_low_ on *A*_high_. However, in a case where greater flexibility is needed, that is GLMs fail to sufficiently__ capture subtle impacts of these predictors, it could be beneficial to explore the broader class of GAMs. Extending our method to use a broader class of GAMs in lieu of GLMs would be relatively straightforward: we would construct the surfaces *S*_ɸlow, Alow_, *S*_Alow_, and *S*_ɸlow_ as before,__ but would replace the models [1]-[3] with GAMs, which could include additional parameters related to the splines. We now mention this important extension in the Discussion section as follows:

“The proposed method can easily be extended by inclusion of additional predictors in the GLM. Polynomial *A*_low_ predictors, rather than the current linear *A*_low_ predictors, may better capture the relationship between *A*_low_ and *A*_high_. […] The code developed to implement the method is flexible and modular, which facilitates modifications and extensions motivated by the particular data analysis scenario. This modular code, available at […]”

2) Presumably, the formula for R is an attempt to mimic the R^2^ metrics used in linear regression. However there already exists of series of pseudo-R^2^ metrics for GLMs, for which pros and cons have been discussed, see e.g. https://web.archive.org/web/20130701052120/http://www.ats.ucla.edu:80/stat/mult_pkg/faq/general/Psuedo_RSquareds.htm. These measures should not suffer from the problem of using an unbounded space as is the case here for R_AAC_ and R_CFC_. The authors should pick one of these and apply it instead of the R measure (unless authors provide a clear explanation of why they used such definition of R).

First, we agree that – in retrospect – the choice of symbol R may be confusing. In this manuscript, the measure R is based on the distance between fitted distributions. This notation is motivated by our previous work in (Kramer and Eden, 2013). Unlike the R^2^ metrics for linear regression, our measure is not meant to estimate the goodness-of-fit of the models to the data, but rather to estimate differences between fits from two different models. To make clear this distinction, we have updated our manuscript to include the following text:

“The statistic R_PAC_ measures the effect of low frequency phase on high frequency amplitude, while accounting for fluctuations in the low frequency amplitude. To compute this statistic, we note that the model in Equation 3 measures the combined effect of 𝐴_low_ and 𝜙_low_ on 𝐴_high_, while the model in Equation 2 measures only the effect of 𝐴_low_ on 𝐴_high_. Hence, to isolate the effect of 𝜙_low_ on 𝐴_high_, while accounting for 𝐴_low_, we compare the difference in fits between the models in Equations 2 and 3.”

“However, in the presence of PAC, we expect 𝑆_𝐴low, 𝜙low_ to deviate from 𝑆_𝐴low_, resulting in a large value of R_PAC_. We note that this measure, unlike R^2^ metrics for linear regression, is not meant to measure the goodness-of-fit of these models to the data, but rather the differences in fits between the two models.”

3) Comparing statistically CFC values between distinct conditions is a key experimental method. The method described in rat data looks really nice, but it would need to be assessed beforehand on synthetic data. Will the method display false alarms (incorrect detection of changes in CFC) when the overall amplitude of the low frequency changes between conditions, as would direct comparison of CFC values between conditions? A method that gets rid of this confound would be a major advancement. Moreover, authors fail to explain the discrepancy in rat data between MI results and GLM results. Is it because of variations of the amplitude of the low frequency signal between conditions?

The reviewer makes an important point: a simulation that mimics the results observed in the rodent data would enhance interpretation of these results. To that end, we now include in the revised manuscript new simulations to mimic the observed results in the rodent data. As recommended by the reviewer, we change the overall amplitude of the low frequency signal (𝐴_low_) and the AAC between two conditions, and compare the MI and R_PAC_ between these two conditions. We find, in the absence of actual PAC, significant MI values while the R_PAC_ values remain insignificant (please see text in comment 8.2).

4) Authors quite surprisingly test their different methods of computing p-values of different synthetic datasets, and never on the same. It is frustrating that one cannot conclude in the end which method provides better results. Moreover, the two methods for generic synthetic data should be presented next to each other to allow comparison (the presentations of the equations could be made more similar).

We agree with the reviewer that, in retrospect, computing p-values in different ways for different simulations was confusing. To address this, we have eliminated the use of analytic p-values in favor of bootstrap p-values. Doing so better aligns our analysis of the simulated data with the in vivo data, and simplifies the analysis presentation. Similarly, in the revised manuscript, we now generate synthetic data with PAC using only one method. Doing so greatly simplifies our analysis and presentation, and additionally circumvents the circularity concern raised in point (8) below.

To make these changes, we have removed the section “Assessing significance of AAC, PAC, and CFC with analytic p-values”, and have renamed the section “Assessing significance of AAC, PAC, and CFC with bootstrap p-values” as “Assessing significance of AAC and PAC with bootstrap p-values”. We note that, in this revised section, we no longer compute a p-value for the CFC. We have chosen to eliminate p-value calculations for CFC to focus on the specific CFC types of interest (i.e., PAC and AAC); doing so further simplifies the manuscript presentation. This section begins:

“Assessing significance of AAC and PAC with bootstrap p-values. To assess whether evidence exists for significant PAC or AAC, we implement a bootstrap procedure to compute p-values as follows. Given two signals 𝑉_low_ and 𝑉_high_, and the resulting estimated statistics R_PAC_ and R_AAC_ we apply the Amplitude Adjusted Fourier Transform (AAFT) algorithm (Siegel et al., 2009) on 𝑉_high_ to generate a surrogate …”

5) It is important to see how the GLM method compares to other PAC detection method (e.g. MI) for weaker modulations. Is it as sensitive? Moreover, reviewers felt that the comparison of the sensitivity of the PAC measures to changes in A_low_ (Figure 5) was quite unfair. MI is an unbounded measure while R cannot be larger than 1. If reviewers understood correctly, the base value for scale factor should be around 0.5 (from Figure 4F), so that it cannot increase more than a twofold increase. This could explain the plateau in Figure 5. Is this correct? Perhaps performing the same analysis with a much weaker coupling would remove this concern.

To address this comment, we have included new simulations to compare the GLM method and the modulation index for weaker modulations. We have updated the manuscript to include the following new subsection “R_PAC_and modulation index are both sensitive to weak modulations”

“To investigate the ability of the proposed method and the modulation index to detect weak coupling between the low frequency phase and high frequency amplitude, we perform the following simulations. For each intensity value 𝐼_PAC_ between 0 and 0.5 (in steps of 0.025), we simulate 1000 signals (see Methods) and compute R_PAC_ and a measure of PAC in common use: the modulation index MI (Theller et al., 1992) (Figure 6). We find that both MI and R_PAC_, while small, increase with 𝐼_PAC_; in this way, both measures are sensitive to small values of 𝐼_PAC_.”

We also note that R is an unbounded measure, as it equals the maximum absolute fractional difference between distributions, which may exceed 1. We now state this clarification in the revised manuscript as follows:

“However, in the presence of PAC, we expect 𝑆_𝐴low, 𝜙low_ to deviate from 𝑆_𝐴low_, resulting in a large value of R_PAC_. We note that this measure, unlike R^2^ metrics for linear regression, is not meant to measure the goodness-of-fit of these models to the data, but rather the differences in fits between the two models. We also note that R_PAC_ is an unbounded measure, as it equals the maximum absolute fractional difference between distributions, which may exceed 1.”

6) The modulation of PAC by the low-frequency amplitude is an important contribution of the paper. It would deserve an actual figure for the second patient data, illustrating it with the method and showing modulations of A_high_ by Ф_low_ splitting the dataset into low and high A_low_. From a mechanistic point of view, the fact that PAC is larger when A_low_ is larger makes perfect sense, and it has been linked to the generation of AAC (Hyafil et al., 2015 Figure 3). Is there an intuitive generative mechanism that would create lower PAC for larger A_low_ (as in Figure 7)?

As recommended by the reviewer, we now include in the revised manuscript a new figure showing the modulation of *A*_high_by *Ф*_low_for the data from the second patient. We show the results of the complete model surface in the three-dimensional space (*Ф*_low_, *A*_low_, *A*_high_), and the components of this surface when *A*_low_ is small, and when *A*_low_ is large. We describe this new figure in subsection “Application to in vivo human seizure data” as follows:

“We show an example 𝑆_𝐴low,𝜙low_ surface, and visualizations of this surface at small and large *A*_low_ values, in Figure 12.”

We agree with the reviewer that linking the simulated and observed CFC to candidate biological mechanisms is an important – and very interesting – goal. However, as suggested by the editor, we refrain from speculating on these generative mechanisms in this methods-focused manuscript.

7) Figure 9 and Figure 10 make apparent that the recovered amplitude of the low frequency signal A_low_ fluctuates within each cycle, which goes against the very notion of amplitude of an oscillation. This can lead to falsely detecting AAC when there is PAC, as is evident for example in Figure 9: A_high_ is higher at the phase of the low-frequency signal where A_low_ is higher. Proper AAC should mean that A_high_ is higher for entire cycles of low frequency where A_low_ is high/low.

The reviewer raises an important issue, which made clear the difficulty in interpreting Figure 9 and Figure 10 of the original manuscript. As noted by the reviewer, in the original Figure 9, the low frequency signal visible in the unfiltered trace (V, blue) was slower than the low frequency band we isolated to study. To address this, we now select a low frequency band (1-3 Hz) more consistent with the dominant rhythms visible in the unfiltered signal. In addition, to allow a more direct comparison between *A*_high_ and *A*_low_, we have updated the figures to include *A*_low_. Finally, we have removed the second example of CFC in human seizure data, to reduce the number of examples and figures in the paper.

8) Clarification of confounding aspect intention and circularity perception in the method:8.1) The simulation part is suboptimal and to some extent based on circularity. Specifically, the method is based on modelling the Hilbert-envelope of high frequency band limited activity with different GLMs, using regressors that are functions of the Hilbert envelope and phase of low frequency band limited activity. The test statistic is derived from the response functions, and statistical inference can be done with parametric methods, or, more appropriately for real data, using bootstrap techniques.For the simulations, a generative model was used which essentially builds the high frequency amplitude component as a function of low frequency phase component and/or the low frequency amplitude component. Thus, it is not really surprising that the method works, specifically if in the processing and generation of the data similar filter passbands etc. have been used. It is unclear how this can be alleviated, unless the perceived circularity is actually not there, or the authors are able to come up with a fairer way to simulate the data in order to convincingly show that their method works in general.

We agree with the reviewer that simulating and measuring PAC with the same generative models weakens the significance of the results. Therefore, we have updated the manuscript to simulate all instances of PAC using the pink noise based method, rather than the GLM-based method. In addition, in the revised manuscript, we now only utilize bootstrap p-values to assess significance. These two changes focus the results on methods applicable to real world data, make the manuscript less verbose, and address the circularity concern.

In the revised manuscript, we now omit the sections Assessing significance of AAC, PAC, and CFC with analytic p-values, and we have revised the section Synthetic Time Series with PAC to reflect our use of the pink noise based method to generate simulated time series as follows:

“We construct synthetic time series to examine the performance of the proposed method as follows. First, we simulate 20 s of pink noise data such that the power spectrum scales as 1⁄𝑓. […] We create a new signal 𝑉′ with the same phase as 𝑉_high_, but with amplitude dependent on the phase of 𝑉_low_ by setting,

𝑉’_high_ = M 𝑉_high_ .

We create the final voltage trace 𝑉 as

𝑉 = 𝑉_low_ +𝑉′_high_ +𝑐∗𝑉_pink_,

where 𝑉_pink_ is a new instance of pink noise multiplied by a small constant 𝑐 = 0.01. ”

8.2) Next, in support of the claim that the proposed test-statistic 'accounts for confounding effects', I feel that the evidence presented at most shows that the R_CFC_ metric scales less strongly with low frequency amplitude (Figure 5), this property should not be oversold.

To address further this important concern, we performed two additional simulations (Figure 7 in the revised manuscript) to compare how R_PAC_ and MI behave under fixed PAC and increased *A*_low_ and AAC. These simulations are motivated in part by the results of thein vivo rodent data. In the first set of simulations, we fix PAC at a non-zero value, and increase *A*_low_ and AAC. We find that both R_PAC_ and MI increase with increased *A*_low_ and AAC, but this increase is much less dramatic for R_PAC_. In the second set of simulations, we consider the absence of PAC, under increased AAC and *A*_low_. We find that MI frequently detects significant PAC while R_PAC_ does not. We include these simulation results in the revised subsection “The proposed method is less affected by fluctuations in low-frequency amplitude and AAC”.

“Increases in low frequency power can increase measures of phase-amplitude coupling, although the underlying PAC remains unchanged (Aru et al., 2016; Cole and Voytek et al., 2017). […] We conclude that in the presence of increased low frequency amplitude and amplitude-amplitude coupling, MI may detect PAC where none exists, while R_PAC_, which accounts for fluctuations in low frequency amplitude, does not.”

8.3) With respect to the application to real data, the abstract mentions that "we illustrate how CFC evolves during seizures and is affected by electrical stimuli". This indeed is illustrated by the data, in that the CFC metric is modulated. Yet, it is questionable (specifically for the human ictal data) that this reflects (patho)physiologically meaningful interactions between band-limited neural signal components. If anything, the increased CFC metric highlights the highly non-sinusoidal nature of the ictal spikes, and demonstrates the generic sensitivity of CFC-measures to the non-sinusoidality of the associated periodic signal components. This is a well-known feature, and the most important confounding interpretational issue in most cross-frequency analyses. Therefore, the high expectations based on the manuscript's title were not met in that respect. This important issue needs to be discussed in more detail, and the title should be adjusted for the sake of the reader's expectation management.

We agree with the reviewer that the non-sinusoidal nature of real brain data is an important confound in CFC analysis; it is an issue that has bothered us for some time (Kramer, Tort and Kopell, 2008). As recommended by the reviewer, we have adjusted the Title to manage better the reader’s expectations: “A statistical framework to assess cross-frequency coupling while accounting for modeled confounding effects”

We also now mention this important issue in the revised manuscript as follows:

“Like all measures of CFC, the proposed method possesses specific limitations. We discuss four limitations here. […] Fourth, we note that the proposed modeling framework assumes appropriate filtering of the data into high and low frequency bands. This filtering step is a fundamental component of CFC analysis, and incorrect filtering may produce spurious or misinterpreted results (Aru et al., 2015; Kramer et al., 2008; Scheffer-Texeira et al., 2013). While the modeling framework proposed here does not directly account for artifacts introduced by filtering, additional predictors (e.g., detections of sharp changes in the unfiltered data) in the model may help mitigate these filtering effects.”

9) Estimating phase and amplitude from Hilbert transform requires the signal to be narrow band. Whereas the 4-7 Hz filtering band for theta phase and amplitude estimation is adequate, the 100-140 Hz filtering band for estimating high gamma phase and amplitude is too wide; it is thus likely that the phase and amplitude estimated this way is not accurate. A way to empirically test whether the filtering band is sufficiently narrow is to plot the real and the imaginary part of the filtered signal in a phase portrait to see whether the trajectory moves around a well-formed center. It is the rotation around this center that allows us to define instantaneous phase and amplitude properly from Hilbert transforms.

The reviewer raises an important point. We agree that using the Hilbert transform to estimate the instantaneous phase is ill-suited for a wide frequency band, and therefore choose this low-frequency band to be narrow. We note that, for the high frequency band, we only estimate the amplitude (and not the phase). We do so motivated by the existing neuroscience literature that utilizes wide, high frequency bands in practice (e.g., Canolty et al., 2006) and advocates for choosing high frequency bands that are wide enough (Aru et al., 2015). We also note that the choice of a wide high frequency band is consistent with the mechanistic explanation that extracellular spikes produce this broadband high frequency activity (Ray and Maunsell, 2011).

To state this clearly in the revised manuscript, we now include the following text:

“However, we note that this method is flexible and not dependent on this choice, and that we select a wide high frequency band consistent with recommendations from the literature (Aru et al., 2015) and the mechanistic explanation that extracellular spikes produce this broadband high frequency activity (Sase et al., 2017). We use the Hilbert transform to compute the analytic signals.…”

Numerous ad hoc assumptions go into Equations (1)-(4). It is difficult to follow the logic behind the method. The beauty of the original CFC measures (e.g., Canolty's or Tort's definition of PAC) is that they are simple and intuitive.

To better describe the logic of the new method, we further explain the reasoning that motivates the method, and include the intuition that the proposed analysis measures the distances between distributions fit to the data. We now include this explanation in the revised manuscript as follows:

“Generalized linear models (GLMs) provide a principled framework to assess CFC (Kramer and Eden, 2013; Osipova et al., 2008; Tort et al., 2007). […] If these models fit the data sufficiently well, then we estimate distances between the modeled surfaces to measure the impact of each predictor. ”

In addition, to enhance the flow of the logic, we have simplified the presentation by: (i) removing the calculation of analytic p-values, (ii) removing the null model, (iii) removing the model based simulations, (iv) making the names of the surfaces more intuitive, (v) eliminating p-value calculations for the CFC surface and focusing instead on the specific couplings of interest, PAC and AAC. By simplifying the presentations, we hope that we have made the manuscript’s logic easier to follow.

The synthetic time series considered here are not biologically realistic. First, spiking neural models should be used to generate such time series. In particular, the model should incorporate the property that gamma and high gamma in some way reflect neural spiking. Second, noise should be added to the synthetic time series to mimic the real world recordings; the amplitude of the noise can be used to assess the influence of signal to noise ratio on the various CFC quantities defined.

To address this concern, we first note that, in the simulated data, we do include a pink noise term to mimic the 1/f distribution of power observed in in vivo field recordings of the voltage. While we do not adjust the noise term directly, we do vary the intensity of PAC and AAC (e.g., Figure 5) which illustrates how the signal to noise ratio impacts these quantities.

In addition, as recommended by the reviewer, we have updated the manuscript to include a simple spiking neural model, and now use it to provide an additional simulation demonstrating the ability of R_PAC_ to detect PAC in the presence of fluctuations in low frequency amplitude, while MI is unable to detect this coupling. We have updated the text to include the following new subsection “A simple stochastic spiking neural model illustrates the utility of the proposed method”:

“In the previous simulations, we created synthetic data without a biophysically principled generative model. Here we consider an alternative simulation strategy with a more direct connection to neural dynamics. […] However, the non-uniform shape of the (𝐴_low_, 𝜙_low_) surface is lost when we fail to account for 𝐴_low_. In this scenario, the distribution of 𝐴_high_ over 𝜙_low_ appears uniform, resulting in a low MI value (Figure 10C).”

For the human seizure data, the authors only evaluated their new measures on the data, but did not compute the commonly applied PAC measures for comparison.

As recommended by the reviewer, we now also apply the modulation index to the human seizure data, and include the results as a new panel in the revised Figure 11 (see Comment 7).

We have updated the Results to describe the inclusion of the modulation index as follows: “Repeating this analysis with the modulation index (Figure 11C), we find qualitatively similar changes in the PAC over the duration of the recording. However, we note that differences do occur. For example, at the segment indicated by the asterisk in Figure 11B, we find large R_AAC_ and an increase in R_PAC_ relative to the prior 20 s time segment, while increases in PAC and AAC remain undetected by MI.”

[Editors' note: further revisions were requested prior to acceptance, as described below.]

1) Title: Another suggestion is that one could use "analysis", rather than "modeling" to have a title of "… confounding analysis effects". This is a suggestion, not required.

As recommended, we have updated the Title to read:

“A statistical framework to assess cross-frequency coupling while accounting for confounding analysis effects”

2) The new Figure 2 is fine. It was thought that the comment regarding GLM vs. GAM was misinterpreted. The point was conceptual rather than algorithmic. The essence of the models described here is to capture a nonlinear mapping of the regressor(s), that is then transformed into the observable using a link function and some observation noise: this is exactly what a GAM is. As described in the textbook by Wood (section 4.2), using basis functions such as splines, a GAM can be transformed into a (penalized) GLM, and then exactly fit using the standard GLM procedure that authors have used. (I believe outfitting is rather an outdated form of fitting GAMs). Please simply add some wording to make this clear. That is, the suggestion is not to change the core model fitting part, only to acknowledge the direct link with GAM.

We thank the reviewer for this clarification. We agree that the models are situated within the class of GAMs. To acknowledge the direct link with GAMs, we have added the following text to the revised manuscript:

“We note that, because the link function of the conditional mean of the response variable *A*_high_varies linearly with the model coefficients 𝛽_k_, the model is a GLM, though the spline basis functions situate the model in the larger class of generalized additive models (GAMs).”

3) We understand that the R measure captures how strongly the regressors modulate high frequency amplitude: this is exactly what R^2^ does in linear regression (assessing the part of the variance explained by the model), although it is sometimes described incorrectly as measuring goodness-of-fit. In their response, reviewers claim a somehow different thing, that their measure "estimate differences between fits from two different models". Now, if this refers to model comparison, again, there is large literature on how to perform model comparison between GLMs (including AIC, cross-validation, etc.). So I can see no reason to create a new method that suffers some important drawbacks (not derived from any apparent principle, arbitrariness for unbounded regressors, PAC measure is modulated by levels of AAC and low frequency amplitude, etc.) and has not been tested against those standard measures. We fully support the idea of using GLM as statistical tools for complex signals; it is a pity though not to leverage on the rich tools that have been developed within the GLM framework (and beyond that in machine learning). Please either use these rich tools or provide a proper argumentation of why your measure is used.

To address this important concern, we have added new results and additional discussion to the revised manuscript. We show that two standard model comparison methods for nested GLMs frequently detect PAC and AAC in pink noise signals. We have added the following text to subsection “The absence of CFC produces no significant detections of coupling”:

“We also applied these simulated signals to assess the performance of two standard model comparison procedures for GLMs. […] We conclude that, in this modeling regime, two deviance-based model comparison procedures for GLMs are less robust measures of significant PAC and AAC.”

We have also added the following text to the Discussion section:

“We chose the statistics R_PAC_ and R_AAC_ for two reasons. […] While many model comparison methods exist – and another method may provide specific advantages – we found that the framework implemented here is sufficiently powerful, interpretable, and robust for real-world neural data analysis.”

4) The new simulations represent an important step in the good direction but the analysis performed on the synthetic data is not quite the one performed on the experimental data. Figure 7 shows that the R measure is modulated by amplitude of the low frequency signal as well as AAC (although less so than the MI measure), preventing any direct comparison of PAC between differing conditions. Now on the rodent data, authors very astutely use a single model for both conditions, and indicator functions (Equation 10). For some reason, the model comparison performed on the previous version of the manuscript has disappeared. This was viewed as regrettable as this indicated that PAC was modulated between the two experimental conditions, something that direct comparison of MI/R values cannot afford to test. This is also the manipulation that we would like to see tested on synthetic data, to make sure whether authors are using a method to isolate modulations of PAC from modulations of low frequency amplitude and AAC.In other words, it would be great to see this method tested on synthetic data to see if they get rid of the confounds, and then possibly applied on rat data as in previous manuscript. This was viewed as something that could make the paper much better. However, it is acceptable to also choose to trim the comparison between conditions part. If so, the authors are requested to be very explicit in their wording that by using separate models for different conditions, they cannot get rid of possible confounds.

We agree that this model comparison is an important result and have updated the manuscript to include this result in subsection “Using the flexibility of GLMs to improve detection of phase-amplitude coupling in vivo” as follows:

“To determine whether the condition has an effect on PAC, we test whether the term

P∑j=1nβn+3+jfjlowin Equation 9 is significant, i.e. whether there is a significant difference between the models in Equations 9 and 10. […] We conclude that this method effectively determines whether stimulation condition significantly changes PAC.”

5) The R measure seems pretty much as sensitive as the MI measure to low PAC. However, the two measures cannot be compared directly. Since most research is geared towards assessing significance levels of PAC, it would be interesting to rather show how the two methods compare in terms of statistical power: comparing type I error rates for low levels of PAC (although reviewers are a bit wary of what could be the results, since the type II error rate related to Figure 5 was less than 1% when it is supposedly calibrated to be 5%, suggesting the method could be too conservative).

In the revised manuscript, we now show how the two methods compare in terms of statistical power. We have updated Figure 5.

We have also updated the results text in subsection “R_PAC_ and modulation index are both sensitive to weak modulations” as follows:

“We find that both MI and R*_PAC_*__, while small, increase with *I**_PAC_*__; in this way, both measures are sensitive to small values of *I**_PAC_*__. However, we note that R_PAC_ is not significant for very small intensity values (*I*_PAC_ <= 0.3), while MI is significant at these small intensities. We conclude that the modulation index may be more sensitive than ****R_PAC_ to weak phase amplitude coupling.”

*6) The authors do not seem to have taken into account a reviewer's response to their revision workplan, so it is repeated here. The response to point 7 misses the important message: the extraction of the amplitude of the low frequency signal is intrinsically flawed as the recovered amplitude fluctuates* within *cycles of the low frequency rhythm. Looking at Figure 9C, peaks of A_low_ and A_high_ coincide within these cycles, which will very likely give rise to spurious AAC for this segment (irrespective of whether slow fluctuations of A_low_ do modulate A_high_). It is unclear why this is apparent here but was not detected in the analysis of synthetic data, but in any case, it would require in our opinion a serious assessment of the properties of the algorithm to extract A_low_. The authors' solution (changing the frequency range of the simulation to make the problem less apparent) is not a valid one. In any case, we can still see the issue in Figure 11F: sub-second fluctuations in A_low_. Perhaps one quick fix would simply be to use a lowpass filter for A_low_ to remove these spurious fluctuations. It makes no sense conceptually that a signal assessing the amplitude of an oscillation would fluctuate rapidly within each cycle of that oscillation.*

We would like to thank the reviewer for pointing out this filtering issue. To address this concern, we re-examined the spectrogram of the full signal (updated Figure 11B), and identified a limited region of increased power in the 4-7 Hz range from 130 s to 140 s. We then focused our analysis only on this time interval, and selected a filter to isolate the 4-7 Hz range. We note that, in our previous analysis, we applied the same filter over the entire duration of the seizure, which – due to the changing dominant rhythms during a seizure – complicated the subsequent analysis. By focusing on a time period with a clear 4-7 Hz rhythm, we greatly improved the estimate of *A*_low_ (Figure 11C). We have updated this subsection “Application to in vivo human seizure data” as follows:

“To evaluate the performance of the proposed method on in vivo data, we first consider an example recording from human cortex during a seizure (see subsection “Human subject data”). […] Comparing *A*_low_ and *A*_high_over the 10 s interval (each smoothed using a 1 s moving average filter and normalized) we observe that both *A*_low_ and *A*_high_steadily increase over the duration of the interval.”

*7) Regarding point 8.1 in the rebuttal: the change proposed in generating the signals was viewed as just a minor cosmetic change. It does not take away our concern for circularity, since the high frequency amplitude is still a direct function of the phase time series of the low frequency signal. Also, a change in signal generation was only done for the PAC, and not for the AAC, which suffers from the same circularity concern. As mentioned before, perhaps this concern cannot be alleviated at all (although we think that the authors could have done a better job by generating both the low frequency phase time series* and *the high frequency amplitude time series as a (possibly non-linear) function of a third unobserved time series. Either way, the authors need to at least discuss this concern for circularity explicitly, and argue why this in their view is not a problem in convincingly showing the utility of their new method.*

Our goal in simulating PAC and AAC was to create time series that explicitly contained these types of CFC. We agree with the reviewer’s point that detecting the simulated PAC and AAC is not surprising; we expect that GLMs are able to capture these relationships. However, we feel that the broad, interdisciplinary audience of this journal would benefit from a demonstration of the efficacy of the analysis framework on data that are known to have PAC and AAC, i.e. on the simulated data. To that end, we simulated data of the size we expect to analyze in real neural systems, and have shown that the proposed method has enough power to detect different types of CFC. This type of demonstration is common for new methods in neural data analysis (and in CFC methods specifically), and simulated PAC data with this particular structure are also common in the neuroscience community (e.g., Lepage and Vijayan, 2015, Section 2.1, Equation 6).

We have updated the revised manuscript to note this point in the Discussion as follows:

“Fifth, we simulate time series with known PAC and AAC, and then test whether the proposed analysis framework detects this coupling. The simulated relationships between *A*_high_ and (*Ф*_low_, *A*_low_) may result in time series with simpler structure than those observed in vivo. For example, a latent signal may drive both *A*_high_ and *Ф*_low_, and in this way establish nonlinear relationships between the two observables *A*_high_ and *Ф*_low_. We note that, if this were the case, the latent signal could also be incorporated in the statistical modeling framework (Widge et al., 2017).”

8) Regarding point 8.3 in the rebuttal: we find the proposed fourth concern far too theoretical to be of practical use. The readership really should be informed about the interpretational confounds of CFC metrics, as mentioned in an original comment. We are not convinced that 'appropriate filtering of the data into high and low frequency components' is fundamentally possible, but we'd like to stand corrected with a convincing argumentation. In other words, the authors are requested to be very explicit in the interpretational limitations of any CFC measure, which is independent of the signal processing that is applied to the data before the measure is computed. This is important to be very clear about since 'non-methods' people may use the method without too much thinking about the potential shortcomings.

To address this concern we have updated the Discussion section to focus more specifically on the impact of non-sinusoidal signals on CFC analysis:

“Fourth, we note that the proposed modeling framework assumes the data contain approximately sinusoidal signals, which have been appropriately isolated for analysis. In general, CFC measures are sensitive to non-sinusoidal signals, which may confound interpretation of cross-frequency analyses (Aru et al., 2015; Cohen and Devachi, 2017; Kramer and Eden, 2013). While the modeling framework proposed here does not directly account for the confounds introduced by non-sinusoidal signals, the inclusion of additional predictors (e.g., detections of sharp changes in the unfiltered data) in the model may help mitigate these effects.”

9) The authors are asked to edit a sentence in their Introduction about cellular mechanisms being ‘well-understood’ for gamma and theta – this was viewed as somewhat arguable for theta. Tort et al. is cited for theta which does not seem to be an appropriate reference for cellular perspectives. A recent paper for cellular mechanisms for theta could be used (Ferguson et al., 2018).

As recommended, we have updated the manuscript to read:

“Although the cellular mechanisms giving rise to some neural rhythms are relatively well understood (e.g. gamma: Likhtik et al., 2013; Wagner et al., 2015; Weiss et al., 2015), the neuronal substrate of CFC itself remains obscure. “

10) Please also consider the following: a) Some of the figures/panels could be placed as figure supplements to avoid distracting the reader away from the main points of the manuscript.

Although we appreciate this recommendation, we would prefer to keep all material in the main manuscript.

b) The figures could be polished a bit more, notably:- Use labels such as 'π2' for phase variables.- Merge single lines panels when possible (e.g. Figure 12B,C).- Improve the readability of the 3D figures (e.g. using meshgrids; perhaps plotting only the full model surface in Figure 5).- Adjust font size.

In the revised manuscript, we have included appropriate labels for phase variables, merged single line panels when possible, included meshgrids for 3D figures, and increased the font size.

- Adjust axis limit and panel size for better readability (for some panels it's hard to see anything beyond noise, e.g. Figure 4, Figure 9B,C, Figure 10B).

In the revised manuscript, we have adjusted the axis limit and panel size to better visualize effects beyond noise.

- Figure 5G,K and O are difficult to read.

We have widened these subfigures to be more readable.

- Figure 11B: are these stacked bars of is R_PAC_ always larger than R_AAC_? In the former case, it makes R_PAC_ difficult to read: plot unstacked bars or curves instead.

In the revised manuscript, we have eliminated this subplot; please see our response to (6) above.

- Figure 11F: use different scales from A_high_ and A_low_ to make A_high_ visible (or normalize signals)

In the revised manuscript, we now normalize the *A*_high_ and *A*_ low_ signals in Figure 11E.__

*c) Names of the models: why name them “ɸ*__*_low_” and “A*_low_
*“rather than PAC and AAC models?*

We use *ɸ*_low_ and *A*_low_ to indicate which signals are modulating A___high_ in the respective models. We decided against__ calling these models AAC and PAC models, as R_PAC_utilizes the *A*___low_and *A*_low_,__
*ɸ*_low_ models (formerly AAC and CFC models), but not the *ɸ*_low_ model (formerly, PAC model). Similarly, R_AAC_ uses the *ɸ*_low_ and *A*_low_, *ɸ*_low_ models (formerly PAC and CFC models), but not the *A*_low_ model (formerly AAC model), which may be confusing to readers.

To further clarify the model names, we have added the following text after the definitions of R_PAC_ and R_AAC::_

R_PAC_ = max[abs[1-S*_A_*_low_/S*_A_*_low_,ɸ_low_ ]],

“i.e. we measure the distance between the *A*_low_ and the *A*_low_,*ɸ*_low_ models.”

R_AAC_ = max[abs[1-S*_ɸ_*_low_ /S*_A_*_low_,*_ɸ_*_low_ ]],

“i.e. we measure the distance between the *ɸ*_low_ and the *A*_low_,*ɸ*_low_ models.”

d) Isn't the constant offset β_0_ missing from equation 1 and 3?

We now include the following sentence in the manuscript for clarification:

“The functions {f_1_ … f_n_} correspond to spline basis functions, with n control points equally spaced between 0 and 2 π, used to approximate *ɸ*_low_. We note that the spline functions sum to 1, and therefore we omit a constant offset term.”

e) Please use semi-columns instead of comas to separate possible values of x (it took me a while to understand this).

In the revised manuscript, we have updated this sentence to read:

“Given a vector of estimated coefficients 𝛽_x_ for x = {*A*_low_; *ɸ*_low_; or *A*_low_,*ɸ*_low_}, we use its estimated…”

f) In the spiking model, specify that the slow oscillation is imposed externally, not generated by the network.

“In this stochastic model, we generate a spike train (*V*_high_) in which an externally imposed signal Vlow modulates the probability of spiking as a function of *A*_low_and *ɸ*_low_.”

g) "at the segment indicated by the asterisk in Figure 11B (…)": there is no asterisk in Figure 11B.

This subfigure is no longer included in the revised manuscript.

h) Figure 11F: "A_high_ increases with A_low_ over time"-> not clear. A_high_ increases over time but it seems that it is higher for intermediate than for high value of A_low_.

In the revised manuscript, we analyze a time segment with a clear relationship between *A*_low_ and *A__*_high_. Please see our__ response to question (6).

i) A concern was raised about whether the frequency band corresponding to the low signal in the human data is described in the text.

Thank you, we have updated the Methods section to read:

“For these data, we analyze the 100-140 Hz and 4-7 Hz frequency bands…”